# Spontaneous single-nucleotide substitutions and microsatellite mutations have distinct distributions of fitness effects

Yevgeniy Plavskin[1,2], Maria Stella de Biase[3,4], Naomi Ziv[1,2¤], Libuše Janská[1,2], Yuan O. Zhu[5,6], David W. Hall[7], Roland F. Schwarz[3,8,9], Daniel Tranchina[2,10], Mark L. Siegal[1,2]*

1 Center for Genomics and Systems Biology, New York University, New York, New York, United States of America, 2 Department of Biology, New York University, New York, New York, United States of America, 3 Berlin Institute for Medical Systems Biology, Max Delbrück Center for Molecular Medicine in the Helmholtz Association, Berlin, Germany, 4 Humboldt-Universität zu Berlin, Department of Biology, Berlin, Germany, 5 Department of Genetics, Stanford University, Stanford, California, United States of America, 6 Department of Biology, Stanford University, Stanford, California, United States of America, 7 Department of Genetics, University of Georgia, Athens, Georgia, United States of America, 8 Institute for Computational Cancer Biology, Center for Integrated Oncology (CIO), Cancer Research Center Cologne Essen (CCCE), Faculty of Medicine and University Hospital Cologne, University of Cologne, Cologne, Germany, 9 Berlin Institute for the Foundations of Learning and Data (BIFOLD), Berlin, Germany, 10 Courant Math Institute, New York University, New York, New York, United States of America

¤ Current address: Shmunis School of Biomedical and Cancer Research, Tel Aviv University, Ramat Aviv, Israel

* mark.siegal@nyu.edu

**Data Availability Statement:** Analysis code and data necessary to replicate these findings can be found on the Open Science Framework: https://doi.org/10.17605/OSF.IO/H4J9F. All sequencing data

## Abstract

The fitness effects of new mutations determine key properties of evolutionary processes. Beneficial mutations drive evolution, yet selection is also shaped by the frequency of small-effect deleterious mutations, whose combined effect can burden otherwise adaptive lineages and alter evolutionary trajectories and outcomes in clonally evolving organisms such as viruses, microbes, and tumors. The small effect sizes of these important mutations have made accurate measurements of their rates difficult. In microbes, assessing the effect of mutations on growth can be especially instructive, as this complex phenotype is closely linked to fitness in clonally evolving organisms. Here, we perform high-throughput time-lapse microscopy on cells from mutation-accumulation strains to precisely infer the distribution of mutational effects on growth rate in the budding yeast, *Saccharomyces cerevisiae*. We show that mutational effects on growth rate are overwhelmingly negative, highly skewed towards very small effect sizes, and frequent enough to suggest that deleterious hitchhikers may impose a significant burden on evolving lineages. By using lines that accumulated mutations in either wild-type or slippage repair-defective backgrounds, we further disentangle the effects of 2 common types of mutations, single-nucleotide substitutions and simple sequence repeat indels, and show that they have distinct effects on yeast growth rate. Although the average effect of a simple sequence repeat mutation is very small (approximately 0.3%), many do alter growth rate, implying that this class of frequent mutations has an important evolutionary impact.

was deposited in the Sequence Read Archive under project PRJNA1117962.

**Funding:** This work was supported by National Institutes of Health grants R35GM118170 and R35GM148344 (to MLS), and National Institutes of Health Grant R01GM097415 (to MLS and DWH). RFS is a Professor at the Cancer Research Center Cologne Essen (CCCE) funded by the Ministry of Culture and Science of the State of North Rhine-Westphalia. This work was partially funded by the German Ministry for Education and Research as BIFOLD - Berlin Institute for the Foundations of Learning and Data (ref. 01IS18025A and ref 01IS18037A to RFS). LJ was supported by a NYU Dean's Undergraduate Research Fund Grant. YOZ was supported by the A*STAR National Science Scholarship PhD. The funders had no role in study design, data collection and analysis, decision to publish, or preparation of the manuscript.

**Competing interests:** MLS is on the Editorial Board of PLOS Biology.

**Abbreviations:** AIC, Akaike information criterion; DME, distribution of individual mutational effect; FDR, false discovery rate; MA, mutation-accumulation; PDF, probability density function; SNM, single-nucleotide mutation; SSR, simple sequence repeat.

## Introduction

Mutations constitute the raw material upon which selection acts. Understanding the properties of new mutations is therefore of central importance to evolutionary biology [1]. For example, the frequency and effect sizes of mutations that increase fitness are key determinants of the rate of evolutionary adaptation [2]. The frequencies of mutations that decrease fitness also impact adaptation, as well as patterns of genetic diversity [3]. In addition, mutational properties are informative of the structure of genetic networks: if a large proportion of mutations affecting a phenotype is non-neutral, then the phenotype can be affected by changes to the function of a large number of genes across the genome, suggesting a high degree of interconnectedness among the gene-regulatory networks operating in the cell. For example, evidence of large numbers of variants affecting complex traits in humans has recently been proposed to support a model of widespread interconnectedness among gene-regulatory networks [4]. The relative contributions of different mutational types (e.g., single-nucleotide substitutions, copy-number variants, repetitive sequence expansions/contractions) to phenotypic differences among organisms is another poorly understood property of mutations. Shedding light on this property is critical not only for understanding a phenotype's propensity to change, but also for selecting appropriate technologies to assay the phenotype's genetic basis [5,6]. Finally, the properties of new mutations are also of interest because of their relevance to human health: de novo mutations are thought to constitute a major set of causative variants for many genetic disorders [7], and the rate of small-effect deleterious mutations has been shown to play a significant role in tumor evolution [8].

Mutation-accumulation (MA) lines in model organisms have allowed unbiased exploration of the properties of new mutations. Repeatedly passaging organisms through extreme bottlenecks for many generations allows mutations to accumulate while largely shielded from selection. The phenotypes of these MA lines can then be assayed, revealing the spectrum of mutational effects of new mutations. Studies have used mutation accumulation to probe mutational effects in diverse organisms, but the resulting estimates of typical effect sizes vary widely, even among studies assaying closely related phenotypes in the same species (reviewed in [9,10]). Two culprits likely explain the discrepancies. First, MA studies have historically lacked genotypic information. That is, it was not known how many mutations were present in each strain, let alone how many trait-altering mutations there were. Many studies addressed this issue by assuming a parametric distribution representing a single mutational effect; each MA strain was then modeled as containing a Poisson random number of mutations with an unknown mean. The parameters of the distribution of mutational effects were then jointly fitted with a parameter representing the mean number of mutations present across the MA strains of interest. However, these estimates of mutation rate are difficult to interpret because in most cases, the confidence intervals of such estimates have no upper bound (see for example [9,11,12]). This problem is caused in part by the second culprit: noisy phenotype measurements. The identification of small mutational effects depends on the amount of measurement noise. In addition, because estimates of mutational parameters are confounded with each other [11], the lack of a precise mutation rate estimate translates into uncertainty in the estimates of the other mutational parameters, which describe the shape of the effect distribution.

Several recent studies have sequenced MA lines to make independent measurements of mutation rate. The expected numbers of mutations per line for these sequencing-based mutation rates tend to far exceed the expected numbers of non-neutral mutations estimated from phenotypic measurements in MA lines. This observation has led to the conclusion that in most cases, the majority of substitutions are neutral or nearly neutral with respect to the observed phenotype (reviewed in [10]). However, caution must be taken in transferring mutation rate

estimates between different MA experiments. There is ample evidence that mutation rate is highly experiment-dependent even within a species, with substitution rates differing with strain ploidy, genetic background, and even the environmental conditions in which the mutation accumulation occurred [13–16]. Recent work using either direct measurement of accumulated mutation number in phenotyped MA lines in *Chlamydomonas reinhardtii* [17], *Drosophila melanogaster* [18], mice [19], and *Escherichia coli* [20] or measuring mutation number and phenotype in parallel MA experiments in a mismatch repair-deficient strain of *E. coli* [21] has provided more precise estimates of the distribution of mutational effect size in these species. For example, Robert and colleagues [21] and Böndel and colleagues [17] both show strong evidence for highly leptokurtic (L-shaped, with most mutations having very small effect sizes) distributions of fitness effects in *E. coli* and *C. reinhardtii*, respectively, and Sane and colleagues [20] identify significant differences in the rate of beneficial mutations between transitions and transversions in *E. coli*.

Interpretation of mean mutational effect sizes is further complicated by the fact that although precise estimates of single-nucleotide mutation (SNM) rate are now available from MA studies across a wide range of organisms and conditions, other frequent mutation types, especially mutations in simple sequence repeats (SSRs), are more difficult to identify using conventional analyses of next-generation sequencing data [22,23]. However, because of their repetitive nature, these regions are particularly prone to acquiring mutations by forming loops during replication (polymerase slippage events), which lead to contraction or expansion of the repeat locus. Recent advances in genome-wide SSR genotyping (e.g., [24]) have allowed high-throughput studies of the effects of SSR variants, which demonstrated that variation in these difficult-to-genotype mutation types contributes significantly to phenotypic variation in nature: thousands of short SSR loci contribute substantially to the variance attributed to common polymorphisms affecting gene expression across human tissues and cell lines [6,25,26], rare variants and de novo mutations in SSRs are associated with autism spectrum disorder [27,28], and expression of genes whose promoters contain these repeats diverges more than SSR-free promoters among closely related yeast species [29]. Evidence that mutations in short repeats may contribute significantly to the spectrum of mutational effects is also emerging in MA studies. For example, it has been suggested that the higher estimate of fitness-altering mutation rate in *Dictyostelium discoideum* when compared to other single-celled organisms may be explained by the large number of SSRs in its genome, and the resulting high frequency of expansion/contraction events occurring at these highly mutable loci [9]. More direct evidence comes from estimating the frequency of SSR mutations in MA experiments in *Daphnia pulex* [30]. Selection against SSR mutations was demonstrated by comparing their prevalence in an MA experiment to a control in which selection was active [30]. However, with the exception of that study, little is known about the relative contribution of SNMs as compared to SSR indels and other mutation types to the full spectrum of mutational effects.

Precise and accurate phenotypic measurements are especially important if mutations of small effect dominate. In microbes, batch culture can be used to generate growth rate measurements averaged across tens of millions of cells within a population. However, such measurements can still have appreciable errors, likely caused by the interactions of small biological and technical variations with the exponential growth process: for example, in one study, yeast growth rate measured by optical density in batch culture varied across replicates with a standard deviation of 3% of the mean [31], limiting the ability to detect small mutational effects to strains with the most extreme effects or largest numbers of mutations. Moreover, in laboratory strains of budding yeast, frequently occurring respiration-deficient, slow-growing "*petite*" cells can stochastically bias average growth rates downwards to extents that are independent of the genetic properties of each individual strain [32,33]. We have developed an alternative to batch

culture measurements that uses time-lapse microscopy to perform growth rate measurements simultaneously in tens of thousands of microbial microcolonies [34,35]. Because of the highly replicated nature of the assay, it yields very precise estimates of strains' mean growth rates [35,36].

Here, we combine sequence information, precise growth rate measurements, and modeling to interrogate the properties of spontaneous mutations in yeast. We are particularly interested in 3 key questions: how frequent are small-effect deleterious mutations, what proportion of the genome affects growth when mutated, and how do the effects of different classes of mutations affect growth? To answer these questions, we estimate the effects of spontaneous mutations on growth rate, a complex phenotype closely related to microbial fitness, in 2 sets of MA lines with different mutation spectra. We first show that our microscopy-based growth rate assay allows us to accurately and precisely estimate the net effect on growth of the mutations in each line, notwithstanding stochastic variation in the proportion of slow-growing petite cells across the experimental samples. We next use these individual-level growth data along with substitution rate data from MA lines to fit a distribution of mutational effects. Our results demonstrate that the distribution of spontaneous SNMs is highly skewed towards mutations with extremely small effects on growth rate, and that the vast majority of these mutations decrease growth rate in rich media. Finally, we use an additional, slippage repair-deficient set of MA lines to show that spontaneous indels in SSRs significantly affect growth rate. By applying high-throughput phenotyping and integrating genotype and phenotype data into a single framework for fitting mutational effects, we show that the effects of spontaneous mutations accumulated in MA experiments can be parsed into multiple classes and that SSR mutations make important contributions to trait variation, on the order of a quarter of the combined effect of SNMs. Our results underscore the role that deleterious load from a range of mutational types is likely to play in clonal evolution.

## Results

### Statistical modeling accounts for across-strain variability in the proportion of respiration-deficient (petite) colonies

Our study seeks to infer the effects of spontaneous mutations on yeast growth rate. However, estimating the growth rates of interest is nontrivial. Laboratory strains of *Saccharomyces cerevisiae* are prone to the spontaneous formation of petites, mutants with impaired mitochondrial function that grow at a slower rate than their non-petite counterparts [32,33]. Variation in petite numbers across samples can arise from chance events that cause different numbers of petites in the original founder populations for each sample. Such variation would impact the mean growth rate estimated in each strain, resulting in estimates that reflect stochastic inter-strain differences in petite proportions, obscuring the genetic effect of mutations on the rate of growth of non-petite cells.

To determine whether differences in petite proportions across MA line estimates could be impacting growth rate estimates, we first tested whether experimental aliquots of genetically similar strains truly differ in the proportion of petite cells. On petri dishes, petite colonies in *ade2* mutant strains can be identified by color, as they lack the red color typical of mutants in the end stages of the adenine biosynthesis pathway [37]. We therefore assayed the proportion of petites in a set of 18 MA lines described in [38]. Because these strains differed from each other by only approximately 2 mutations on average, large variation in the proportion of petites across these strains was not likely to be explained by genetic differences among the strains. The line with the highest proportion of non-red colonies had a large proportion of non-petite (large, rapidly growing) colonies that were not red, indicating a decoupling between

colony color and respiratory ability; this line was excluded from this analysis. We found significant variation in the proportion of colonies that were red across the remaining lines ($p \ll$ 0.001 by likelihood ratio test, see Methods) (S1A Fig).

We next sought to determine whether we could accurately estimate the petite proportion in each strain directly from microcolony growth rate data. Unlike batch culture-based measures of growth rate, which estimate population-average growth rates, the output of the microcolony growth rate measurements we performed is a distribution of individual microcolony growth rates for each sample (Fig 1A) [35,39]. We therefore can make estimates of the proportion of petites directly from microcolony growth rate data, while simultaneously estimating the mean

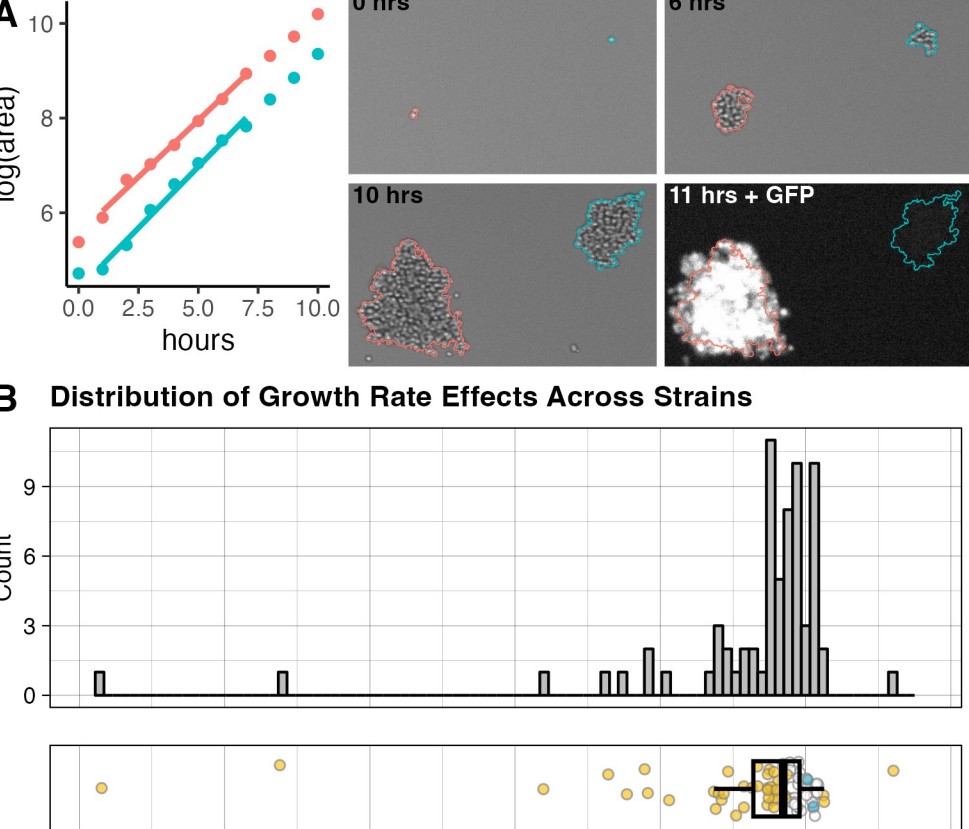

**Fig 1. Measuring the cumulative effects of spontaneously accumulated mutations on growth rate across MA strains.** (A) The microcolony growth rate assay. Microcolony growth rates are measured in parallel using automated microscopy and image analysis. Fluorescent imaging at the end of the growth period is used to differentiate between the MA strain and the ancestor-derived strain grown in each well as a reference. The brightfield images of 3 time points show automated colony detection for 2 representative colonies: from a GFP-marked ancestral reference (red) and an MA line (blue), growing side-by-side with different starting sizes and lag times, but at similar rates. A fluorescence image taken after the final time point shows the GFP expression in the reference colony. The points on the plot represent log(area) over a 10-h time period measured for these colonies, with the best fit used to determine growth rates for each colony (using a 7-time point window) shown as a line. (B) Mutation effects (*s*) in MA lines relative to ancestral reference strain. Points in the plot on the bottom are colored yellow if their *s* value differs significantly from the ancestor at an FDR of 0.05. Blue points represent mutational effects calculated for 2 control strains derived from independent haploid spores of the ancestral diploid (these are not included in histogram or boxplot calculation). Boxplot shows the 25th and 75th percentiles, and median, of the *s* value across all MA strains. The data and code needed to generate this figure can be found on OSF: https://doi.org/10.17605/OSF.IO/H4J9F. FDR, false discovery rate; MA, mutation-accumulation.

growth rate of the non-petite microcolonies. We model the distribution of colony-wise growth rates as a mixture of 2 Gaussian growth rate distributions, with the parameters of the distribution of petite growth rates estimated from independent petite strains derived from the MA ancestor (see Materials and methods). We found that microcolony assay-derived petite proportions are highly correlated with colony color-based proportion estimates (Pearson correlation coefficient = 0.83), indicating that microcolony growth rate data can be used directly to partition growth rates of petite and respiring colonies. Microcolony assay-based petite proportion estimates are approximately 4% lower than the colony color-based estimates (**S1B Fig**). This discrepancy may be a consequence of underestimation of the proportion of petites using growth rate data; alternatively, the discrepancy may arise as a result of a small number of non-petite white-colored colonies (which we have seen in these strains [40]). However, because measurements of mutational effect on growth rate are relative to the ancestral strain, a consistent offset in the estimated proportion of petites would result in consistent bias in mean growth rate estimates of all strains, including the ancestor, resulting in accurate estimates of relative growth rates.

## Accumulated mutations have primarily negative effects on haploid growth rate

To assess the effects of spontaneous mutations, we first examined the distribution of growth rates of a set of haploid MA lines, each likely harboring a unique set of mutations. These lines are derived from diploid parent strains that accumulated mutations over the course of 2,000 generations [31,40,41], accumulating an average of approximately 8 SNMs each [42] (or approximately 4 SNMs per haploid strain assayed here). Hall and colleagues reported that 2.5% of these diploid yeast MA strains, and 14% of viable haploid progeny of these strains, had a significant increase in growth rate as compared to the ancestral line [43]. Sequencing revealed that about a fifth of these diploid MA lines harbored aneuploidies [42]; we excluded the progeny of these aneuploid strains from our study to avoid confounding the effects of smaller mutations, although we did not eliminate strains that underwent aneuploidization events during meiosis. We assayed the growth rates of 70 haploid viable MA line progeny (each derived from a single unique diploid parent MA strain) using the microcolony growth assay [34,35,39]. Cells of each MA strain were grown and imaged in independent wells of 96-well plates; each well also contained cells of a reference strain (a GFP-marked haploid line derived from the MA ancestor) to control for well effects on growth (**Fig 1A**). The use of haploid MA lines allows us to assay the effects of any mutations in these lines in the absence of dominance effects.

We are interested in the distribution of the changes in growth rate among MA-derived haploid strains relative to the growth rate of the ancestral strain. We measure these differences as the selection coefficient, *s*, which is positive when mutations are beneficial (increase growth rate) and negative when they are deleterious:

$$s_{MA} = \frac{g_{MA}}{g_{anc}} - 1, \tag{1}$$

where $s_{MA}$ is the selection coefficient representing the combined effect of the mutations in a given MA line on growth rate, and $g_{MA}$ and $g_{anc}$ are the growth rates of the MA and ancestral strains, respectively. We estimate the proportion of petites in each strain by fitting a mixture of Gaussians as described above (and in Materials and methods), and estimating $g_{MA}$ and $g_{anc}$ as the respective means of the non-petite colony growth rates for the MA and ancestral strains.

Before examining the distribution of MA-line selection coefficients, we first tested the effect of partitioning petite and non-petite growth rates. As expected, modeling colony growth rates

in each strain as a mixture of 2 Gaussians that allows for a subpopulation of petites produces a significantly better fit to the data ($p \ll 0.001$ by likelihood ratio test; see Methods) than simply fitting a single Gaussian distribution to the colony growth rates of each MA line (S1B Fig). In addition, the strains subjected to growth rate assays included 2 non-GFP marked control strains, derived from independent haploid spores of the ancestral diploid; these served as independent controls in the experiment, as their growth rates should be the same as that of the ancestral reference strain. As expected, the confidence interval for the selection coefficient $s$ estimated for each of these strains overlaps 0, indicating that they do not significantly differ in growth rate from the ancestral strain. However, these strains do differ from the reference strain in the proportion of petite colonies as estimated by our modeling. As a result, if mutational effects are estimated without accounting for petites, these ancestral control strains are incorrectly estimated to have a significant mutational effect relative to the GFP-marked ancestral control (S1C Fig). Together, these results support the importance of using modeling to separate the effects of stochastically variable petite proportions across strains from the genetic effects of spontaneous mutations on growth rate.

The distribution of MA-line mutational effects (Fig 1B and S1 Table) reveals that the majority of strains contain at least 1 mutation that alters growth rate, and that mutations tend to be deleterious. At a false discovery rate (FDR) of 0.05, approximately 4% of strains have a significant increase in non-petite growth rate (positive $s$ value), and 56% have a significant decrease in non-petite growth rate (negative $s$ value) relative to the ancestral strain. The growth-rate differences tend to be small, with 37% of strains having an s value between 0.01 and -0.05 (1% to 5% decrease relative to the ancestral growth rate), and only 10% of strains having an s value below -0.05 (growth rate decrease below 5%); an additional 9% of strains have significant decreases but with an $s$ value above –0.01. Only a single strain has an $s$ value of $>0.01$.

## SNMs do not fully explain observed mutational effects

One likely source of variation in $s$ values across strains is differences in the effect of these mutations on protein-coding genes. To test whether strains' $s$ values were explained by the predicted severity of the substitutions found in these strains, we sequenced the haploid MA strains and identified SNMs and short indels in non-repetitive regions relative to the ancestral strain, as described in [38]. We identified a total of 307 SNMs and 3 indels across 68 strains. In some cases, multiple nearby SNMs comprised complex mutations in a single locus; by grouping together mutations within 50 bp of each other, we identified 271 mutated loci (S2 Table).

We then predicted the putative effect of each mutation using snpEff [44]. snpEff categorizes each mutation into one of 4 groups: "high" effect mutations, such as nonsense mutations and frameshifts; "moderate" effect mutations, such as in-frame indels and nonsynonymous substitutions; "low" effect mutations, such as synonymous substitutions; and "modifier" mutations, such as mutations outside the coding region of genes. We identified mutations in 262 unique genes (with a small number of genes mutated in more than 1 strain). Among the mutant genes in each strain, 8 had at least 1 mutation with an effect categorized as "high"-impact, 148 had no more than "moderate" effect mutations, and 47 and 65 had mutations predicted to be no more severe than "low" and "modifier" effects, respectively (S2 Table). We also identified aneuploidies in 2 strains (S1 Table); both these strains also had additional mutations. Six strains also lacked any non-repeat mutations.

Because many of our strains contain mutations in multiple loci, we first grouped mutations in each strain and identified the most high-impact mutation that each strain contained. We then compared the magnitude (absolute value) of $s$ values across strains in which the most severe mutations had high, moderate, low, or modifier effects, as well as $s$ value magnitudes in

strains with no mutations (**S2 Fig**). Note that this analysis does not take into account the total number of mutations found in each strain. Although median $s$ magnitudes were higher for strains that included at least 1 moderate- or high-impact mutation, there was no significant effect of the most severe impact type on mean $s$ value (Kruskal–Wallis test $p$-value = 0.35). Critically, four of the 6 strains that did not have any identified mutations had significant growth defects, including 1 strain with an $s$ value of −0.047. This finding indicates that the mutations identified outside of repeat regions in these strains do not fully explain the variation in MA strain $s$ values, and strongly suggests that additional, unidentified mutations are affecting yeast growth rate.

## Modeling reveals distinct distributions of the effects of SNMs and unidentified mutations

We next sought to determine the properties of the distribution of individual mutational effects (DMEs) whose combined effects were observed in **Fig 1B**. To model the DME, we expanded on the approach proposed by Keightley [11]; in short, individual mutational effects are modeled as drawn from a reflected gamma distribution, with sides weighted to represent the different proportions of mutations with positive versus negative effects on the phenotype of interest. The gamma distribution is advantageous because it captures a range of distribution shapes, from highly peaked to exponential, with only 2 parameters: here, we use the mean ($m$) and shape ($k$) of the distribution. To account for the fact that mutations may be biased in the direction of their effects, the 2 sides of the reflected gamma distribution are weighted based on $q$, a parameter representing the proportion of mutations causing a positive effect on the observed phenotype (see http://shiny.bio.nyu.edu/ms4131/MAmodel/ to interactively explore how changes to parameters affect the distribution of mutational effects in MA lines). We treat individual mutations as additive: the net mutational effect in each strain ($s_{MA}$ from **Eq 1**) is the sum of the mutational effects of individual mutations found in that strain. Unlike earlier work, where the number of mutations per strain was not known, here we leverage sequence information to constrain the model. The mean number of non-neutral mutations per strain, $U$, is modeled as half the average number of mutations in the MA strains' diploid parents [42], corrected with a fitted parameter ($p_0$) estimating the total proportion of mutations that are neutral with respect to growth rate (**Eq 13**) (note that $U$ has also been used to denote the deleterious rate specifically [45], which here would be $(1-q)U$). The distribution of observed mutational effects in the MA strains, $s_{MA}$, is therefore modeled as a multifold convolution of the distribution of individual mutational effects.

Although we expected that constraining the model by the known number of mutations per diploid-parent strain would improve fitting, the existence of growth defects in strains lacking identified mutations suggests that there may be a substantial number of mutations missed in the initial sequence analysis of the MA lines whose haploid derivatives are phenotyped here. In particular, the analysis in [42] and the analysis described in the previous section disregarded any repetitive regions, including SSRs, which have a higher mutation rate than the surrounding genome [46]. As a result, the true number of mutations in the MA lines may be the sum of the number of known mutations (almost all SNMs), and of an additional set of "unidentified" mutations, which would include mutations in SSR regions. Therefore, in addition to the "*SNMs only*" model described above, we considered 3 approaches to modeling the distribution of the effects of "unidentified" mutations: the "*single DME*" model, in which both substitution effects and unidentified mutation effects are modeled as being drawn from a single distribution of mutational effects; the "*two-gamma*" model, in which the effects of unidentified mutations are modeled as being drawn from a separate reflected gamma distribution; and a

"*Gaussian*" model, in which substitution effects are modeled as a reflected gamma distribution and the combined effects of unidentified mutations in each strain are modeled as a Gaussian distribution. Below, we lay out the properties and justifications for each of these models in more detail, and then present the results of fitting these models to our data.

If there is no fundamental difference in the distribution of effects of "unidentified" mutations and the distribution of SNM effects, it should be possible to model their effects by releasing the constraint on the average number of mutations per strain (essentially the model proposed by Keightley [11], with no constraint on the value of *U*); the difference between the estimate of *U* in this model and the estimate of non-neutral mutations estimated by our SNM-only model would provide an estimate for the typical number of unidentified mutations per strain. We fit this model to our data in the "*single DME*" model.

The other 2 approaches for modeling unidentified mutational effects are rooted in the possibility that SNMs and unidentified mutations have distinct distributions of phenotypic effects and that our phenotyping data are precise enough to be able to distinguish these 2 distributions. In this case, the effects of SNMs are described as above in the "*SNM-only*" model, but the DME for unidentified mutations is modeled separately in one of 2 ways. First, it is possible to model the effects of these mutations as a reflected gamma distribution with an unknown number of mutations (the "*two-gamma*" model). This is the same model described above for SNMs, with the proportion of positive versus negative mutational effects, the shape and mean of the gamma distribution, and (unlike for SNMs) the average number of unidentified mutations with an effect on growth rate all fitted by the model. However, we hypothesized that the parameter estimates from this model would not be very informative due to the confounding between mutation number and mutation effect size/distribution shape when the total number of mutations is unknown, especially because the effects being modeled by this distribution represent an unknown portion of the total observed effects and the rate of non-neutral mutations must be high enough to be consistent with most strains' differing in growth rate from the ancestral strain.

Considering the lack of information about the number of unidentified mutations in each MA strain, we can instead seek to understand the typical combined per-strain contribution of these mutations. To do so, we modeled the combined effects of unidentified mutations in each line as being drawn from a Gaussian distribution with mean $\mu_{unid}$ and standard deviation $\sigma_{unid}$ ("*Gaussian*" model). In this model, the $\sigma_{unid}$ term fits variance not explained by experimental noise or by the distribution of mutational effects fit to SNMs. Although this model is not informative regarding the parameters of the distribution of single unidentified mutations, it provides useful information regarding the distribution of the cumulative effects of the unidentified mutations on the growth of each MA strain.

In all 3 cases, the observed growth rate of each MA line is the result of the sum of the effects of its SNMs (whose average number per line is known), its unidentified mutations (whose number is unknown), and experimental noise.

We initially fit all 3 models, as well as the "*SNM-only*" model that includes only the effects of sequenced substitution mutations, to the mutational effects estimated for each strain (see Materials and methods for maximum-likelihood estimation procedure, **S1 Table** for the data that was used as input into the models, and **S3 Table** for model results). The Akaike information criterion (AIC) score was lowest for the "*Gaussian*" model, suggesting that this model fits the data best, and that the fit of the "*two-gamma*" model was not sufficiently improved to warrant the addition of the extra parameters. We also found that, as expected, it was impossible to interpret the mutational parameter estimates in the "*two-gamma*" model, which has large confidence intervals; this is likely the result of confounding effects among all parameter values when attempting to fit a distribution of individual mutational effects with an unknown total

mutation number. Importantly, the significantly improved fit of the "*Gaussian*" model over both the "*SNM-only*" and "*single-DME*" models indicates that our phenotypic data were precise enough to identify distinct distributions of the effects of SNMs and unidentified mutations. The better fit of the "*Gaussian*" model relative to the "*single DME*" model in particular implies that the effect distributions of the 2 mutation classes are distinct.

Although our approach of fitting the distribution of mutational effects based on summary statistics of the mutational effects observed in each strain is computationally efficient, it treats uncertainty in the mutational effect estimates for the different strains as uncorrelated; however, in practice, these estimates depend on a number of shared parameters, such as the estimate for the means and standard deviations in growth rate of the reference strain and of petite yeast microcolonies. These dependencies mean the parameter space is likely more constrained than it appears when fitting the model to uncorrelated mutational effect estimates: for example, an overestimate of the mean growth rate of the reference strain would lead to a consistent overestimate of the magnitude of $s$ across all slow-growing MA strains. The correlated uncertainty in strain estimates should propagate to the estimates of DME parameters (in this example, likely leading to an overestimate of the mean effect size of a single mutation). Failing to account for the correlated structure of strain estimates can lead to incorrect estimates of uncertainty on DME parameters and of the relative goodness of fits of different models. We therefore repeated the fit to the distribution of mutational effects model using the microcolony growth rate data directly. We limited this analysis to the "*SNM-only*" model and the "*Gaussian*" model, which had the best fit to the summary statistic-based data. Parameter estimates and confidence intervals were very similar to those estimated in the summary statistic-based fit, with slightly less uncertainty in the parameter estimates of the "*SNM-only*" model when using the microcolony data directly (**Tables 1** and **S3**). Consistent with our previous finding, the "*Gaussian*" model, which modeled SNMs and unidentified mutations as having 2 independent DMEs, provided the best fit to the data ($\Delta$AIC = −10.8, LRT-based $p$ = 0.00002 as compared to the "*SNM-only*" model) (**Table 1** and **Figs 2A** and **S3**).

We find that the vast majority of non-neutral SNMs are deleterious. We further find that the inferred distribution of SNM effects is highly skewed towards mutations with an effect size approaching 0 (**Fig 2B**). As a result, there is large uncertainty regarding the proportion of SNMs that are completely neutral with respect to growth rate; however, at a selection coefficient cutoff of $10^{-6}$ (larger than the reciprocal of effective population size for wild yeast populations, which has been estimated to be on the order of $3.4 \times 10^6$ [47]) our best-fit model indicates that 3% of all substitutions have a significant positive effect on growth rate, and 39% have a significant negative effect on growth rate. Our model estimates that the mean effect of unidentified mutations across the MA lines is likely to be moderately deleterious, and that the typical combined effect of all the unidentified mutations in an MA line is comparable to the effect of a single SNM.

## Spontaneous simple sequence repeat mutations significantly affect growth rate in yeast

Our modeling and sequencing results strongly suggest that SNMs alone do not account for the full range of phenotypic effects observed in the yeast MA lines tested; likely candidates for the source of these mutational effects are SSR loci. To directly test whether mutations in these loci significantly affect growth rate, we measured growth in a set of MA lines mutant for the *MSH3* gene involved in slippage repair [38]. These strains accumulated mutations over the course of 200 generations and contain an average of 1.8 SSR mutations each, the majority of which are deletions of a single repeat unit.

**Table 1. Properties of DMEs identified by alternative models on full data.** A model that accounts for unidentified mutations by fitting a Gaussian distribution representing the effects of these mutations across strains performs better than a model that only accounts for SNMs. Parameter values for each model shown with 95% confidence intervals; ΔAIC is calculated relative to the "SNMs only" model.

| Model | Parameter | | ln(L) | AIC | ΔAIC |
|---|---|---|---|---|---|
| **SNMs only** | $k_{SNM}$ (shape parameter of gamma distribution for SNMs) | **$1.4 \times 10^{-1}$** ($7.8 \times 10^{-2} -$ $4.9 \times 10^{-1}$) | 384,760.6 | −769,359.3 | – |
| | $m_{SNM}$ (absolute value of gamma distribution mean for SNMs) | **$5.0 \times 10^{-3}$** ($3.4 \times 10^{-3} -$ $1.4 \times 10^{-2}$) | | | |
| | $q_{SNM}$ (proportion of SNMs with positive effect on growth) | **10.5%** (2.5%– 24%) | | | |
| | $p_{0\ SNM}$ (proportion of SNMs with no effect on growth) | **0.1%** (0%– 63.1%) | | | |
| **SNMs + unid,"Gaussian"** | $k_{SNM}$ (shape parameter of gamma distribution for SNMs) | **$5.5 \times 10^{-2}$** ($2.0 \times 10^{-2}$–1.8) | 384,768.1 | −769,370.1 | −10.8 |
| | $m_{SNM}$ (absolute value of gamma distribution mean for SNMs) | **$4.1 \times 10^{-3}$** ($2.0 \times 10^{-3} -$ $5.5 \times 10^{-2}$) | | | |
| | $q_{SNM}$ (proportion of SNMs with positive effect on growth) | **6%** (0.9%– 26.6%) | | | |
| | $p_{0\ SNM}$ (proportion of SNMs with no effect on growth) | **5.9%** (0%– 94%) | | | |
| | $\mu_{unid}$ (mean combined effect size of all unidentified mutations within a strain) | **$-3.3 \times 10^{-3}$** ($-7.0 \times 10^{-3} -$ $7.7 \times 10^{-4}$) | | | |
| | $\sigma_{unid}$ (standard deviation of combined effect sizes of all unidentified mutations across strains) | **$5.5 \times 10^{-3}$** ($2.7 \times 10^{-3} -$ $8.6 \times 10^{-3}$) | | | |

AIC, Akaike information criterion; DME, distribution of individual mutational effect; SNM, single-nucleotide mutation.

We first selected *msh3Δ* mutant MA lines for which we could confidently report the absence of any substitutions outside SSR regions, and where any phenotypic change could therefore be attributed to "unidentified" mutations. After 200 generations of MA, only 15 high-coverage sequenced strains reported in [38] had no known substitutions. Samples of the strains had been frozen every 20 generations throughout the mutation accumulation. We therefore genotyped a subset of strains that had high sequencing coverage, but harbored known substitutions, at the 100-generation point. This analysis resulted in an additional four 100-generation *msh3Δ* MA strains with high coverage and no known substitutions. We selected eighteen 200- or 100-generation MA strains without known substitutions for growth-rate analysis (**S4 Table**). It is possible that our chosen strains contain an SNM that was missed. However, based on the previously calculated non-SSR substitution rate in these strains (0.004 substitutions/strain/generation) [38], as well as the proportion of each strain's genome that was either in a repetitive sequence or not sequenced at 10× coverage, we calculate that the total expected number of unidentified non-SSR substitutions across all the phenotyped strains is only approximately 0.6 mutations. As a result, any significant deviations from the ancestral growth rate in these strains are most likely attributable to the effects of SSR mutations.

Our phenotyping of *msh3Δ* MA lines shows that they include many lines with significant deviations from the ancestor in non-petite growth rate (**Fig 3** and **S4 Table**), with 17% of strains having a significant increase in growth rate, and 44% having a significant decrease in growth rate relative to the ancestral *msh3Δ* strain. Strains with significant mutational effects are found both among the strains that accumulated mutations for 100 generations (3 of 4

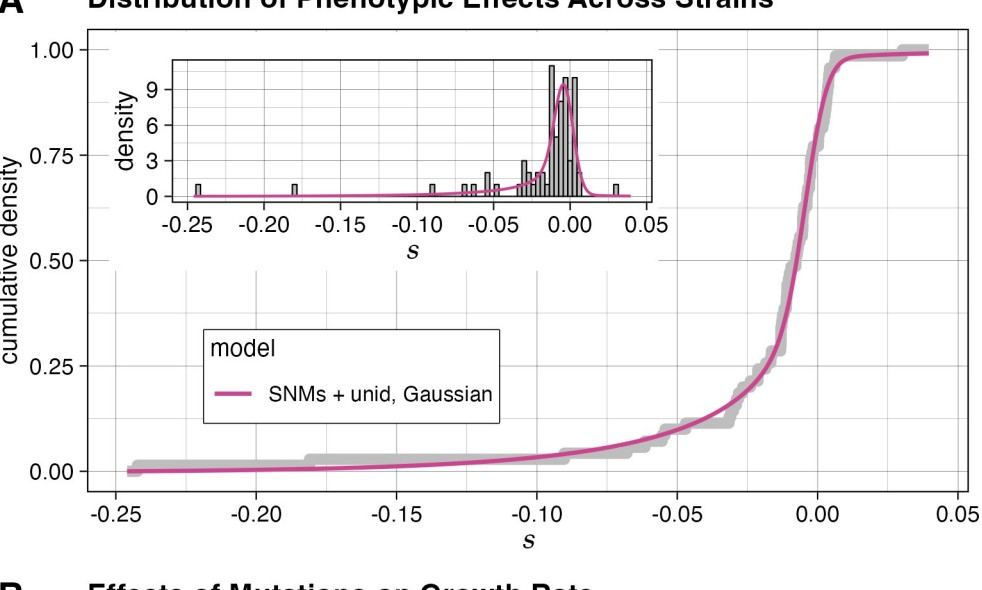

### A  Distribution of Phenotypic Effects Across Strains

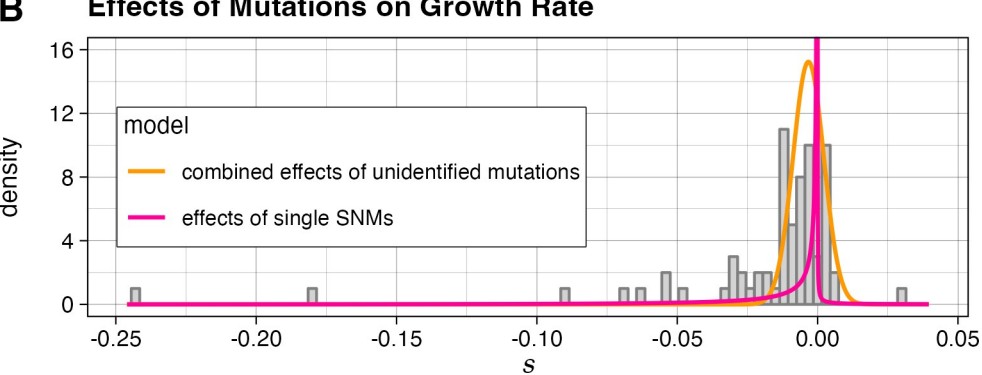

### B  Effects of Mutations on Growth Rate

**Fig 2. The distributions of mutational effects estimated by a model with independent distributions for sequenced and unidentified mutations.** (A) The cumulative density function of the fit of the "*Gaussian*" model (which models SNM effects as a reflected gamma distribution and the sum of all unidentified mutations in a strain as a Normal distribution) to all individual MA strain mutational effects *s*. Inset: histogram of mutational effects, with PDF of the model overlaid. To account for the effect of experimental noise on the estimates of *s*, the model density function is shown convolved with a Gaussian noise kernel with a variance that is the mean of the error variances of each strain's mutational effect estimate. (B) The distributions corresponding to the maximum likelihood estimates of individual effects of SNMs (pink line) and combined effects of unidentified mutations per strain (orange line) plotted over the distribution of MA strain mutational effects. The data and code needed to generate this figure can be found on OSF: https://doi.org/10.17605/OSF.IO/H4J9F. MA, mutation-accumulation; PDF, probability density function; SNM, single-nucleotide mutation.

strains) and among strains that accumulated mutations for 200 generations (8 of 14 strains). The mean mutational effect across all strains was a decrease in *s* of approximately 0.0024 per 100 generations. Considering the fact that there are approximately 1.8 SSR mutations per 200-generation *msh3Δ* MA strain [38], this change corresponds to approximately 0.3% mean decrease in growth rate per SSR mutation. Importantly, the low average number of mutations in each strain, combined with the high proportion of strains with a detected mutational effect on growth rate, suggests that many SSR mutations (at least one third of them) are not neutral.

We next sought to identify potential SSR mutations that may be responsible for the observed effects. In our previous work, we developed an approach to identify a high-confidence, unbiased set of SSR mutations using high-depth sequencing data [38]. We first analyzed

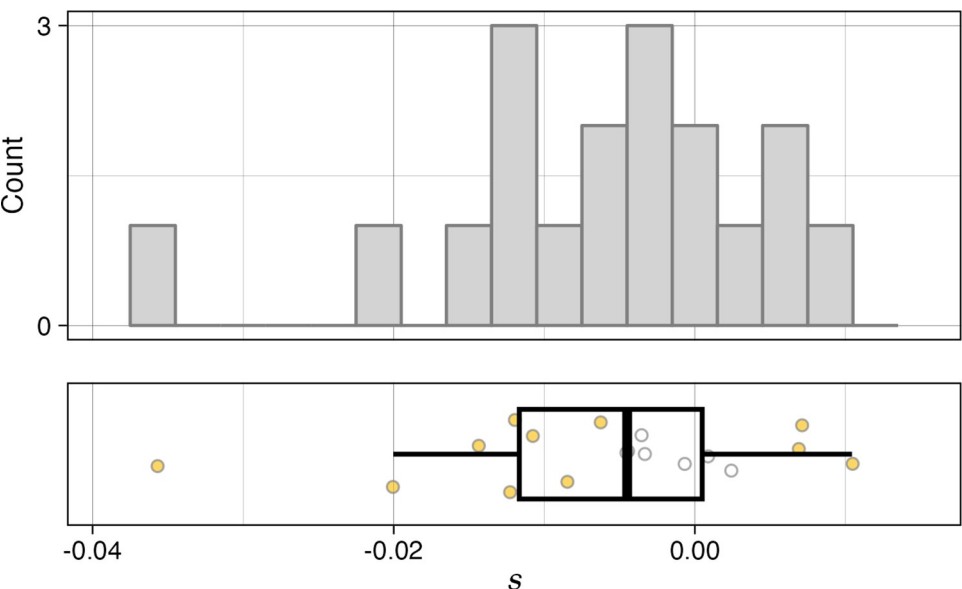

**Fig 3. Mutational effects of SSR mutations.** Mutation effects (*s*) of *msh3Δ* MA lines relative to their *msh3Δ* ancestor. Points in plot on the bottom are colored yellow if their *s* value differs significantly from the ancestor at an FDR of 0.05. The data and code needed to generate this figure can be found on OSF: https://doi.org/10.17605/OSF.IO/H4J9F. FDR, false discovery rate; MA, mutation-accumulation; SSR, simple sequence repeat.

the SSR mutations identified in the *msh3Δ* strains. We identified 6 genes with a "high"-impact mutation, 9 with at least a "moderate"-impact mutation, and 9 with "modifier" mutations; none of the genes had "low"-effect mutations (**S5 Table**); 8 *msh3Δ* MA strains had no identified mutation. Many of the *msh3Δ* MA strains with significant *s* values did not have any identified SSR mutations (**S4 Fig**), and there was no significant effect of the most severe impact type on mean *s* value (Kruskal–Wallis test *p*-value = 0.47); neither finding is unexpected considering the relatively low number of SSRs that we are able to call in these strains [38].

We also performed the analysis from [38] in order to identify a subset of the SSR mutations in the 2,000-generation MA strains used in this study. Of the 49 SSR mutations identified, 14 are deletions, 6 are insertions, 28 are substitutions, and 1 is a complex repeat expansion/substitution. The majority of the mutations (37/49) are in homopolymers (**S6 Table**). About half (23/49) of the SSR mutations were predicted to be modifiers, and 10, 15, and 1 were predicted to have "high," "moderate," and "low" effects, respectively; 35 of 68 strains had no identified SSR mutations. Although this is an incomplete catalog of SSR mutations in these strains, it allowed us to assess whether SSRs may account for the growth rate effects observed in strains lacking non-repeat mutations. Of the 6 strains without SNMs, we identified SSR mutations in 4. Three of these 4 strains had significantly negative *s* values. The strain with the highest-magnitude *s* value among those that lacked non-repeat mutations harbored a single identified SSR mutation with a "high" snpEff-predicted impact. Together, these data provide additional support for our finding that mutations in SSRs contribute significantly to the distribution of fitness effects of de novo mutations.

## Discussion

The distribution of mutational effects determines key properties of a trait's evolution. We sought to understand 3 critical aspects of the characteristics of new mutations: the frequency of deleterious mutations that cannot be effectively purged from adaptive lineages during clonal

evolution; the proportion of the genome that affects growth when mutated, which may be a proxy for the interconnectedness of genetic networks in the cell; and the contributions of different mutation types to growth effects, an important consideration for future approaches to studying evolutionarily relevant genetic diversity. We used MA lines of the budding yeast *S. cerevisiae* to estimate the distribution of the effects of spontaneous mutations on growth, a complex phenotype that contributes substantially to microbial fitness. By combining precise measurements of growth phenotypes with genotypic information, we found that SNMs do not account for all the observed mutational effects, and that an additional set of mutations outside of this commonly investigated mutational class, indels in SSR loci, makes a significant (but smaller) contribution to mutational effects on growth rate. We also found that the distribution of spontaneous substitution effects in yeast is highly skewed towards extremely small effects, consistent with other recent estimates of mutational effects performed across a range of organisms (e.g., [17,21]).

Our work provides multiple lines of evidence that a class of frequent mutations besides single-nucleotide substitutions has significant effects on growth phenotypes in yeast. Our modeling of 2,000-generation MA phenotypic effects suggests that the unidentified mutations whose effects we are detecting have typical combined effect sizes per strain similar to the average effect size of a single substitution. The exact estimate of the contribution of these mutations may be affected by the choice of the parametric distribution of mutational effects used in the model. However, this result matches closely with the mean effect of non-substitution mutations measured directly in the 200-generation *msh3Δ* slippage-repair-deficient mutation accumulation. Previous work in yeast measured the effects of mutations caused by EMS (predominantly SNMs) or in *msh2*-mutant cells (predominantly homopolymer slippage mutations) [48]. That study hinted at the possibility that small-effect mutations might be very common, yet it was admittedly underpowered to detect very small effects, let alone differences between SNM and SSR-mutation effect distributions, and it did not examine the locations of these mutations [48].

Mutations in SSRs are the most likely candidate for the identity of the non-substitution mutations underlying the observed growth rate effects in the 2,000-generation and *msh3Δ* MA experiments. Our previous estimates suggest that in the 200-generation *msh3Δ* MA experiment, SSRs are mutated at a rate of approximately 0.9 loci per genome per 100 generations [38]. These mutations are likely relatively frequent in wild-type strains as well: although estimates of the SSR spontaneous mutation rate vary widely, our work suggests that it may be up to half of the rate observed in the *msh3Δ* background [38]. However, SSRs were masked in the original sequencing analysis of the 2,000-generation MA lines used here [42], as is commonly done due to the difficulty of accurately identifying SSR mutations using conventional variant-calling approaches [24]. Although other mutation types, such as large chromosomal rearrangements and aneuploidies, are likely to have strong phenotypic effects, segmental duplications are very rare: rates in diploid yeast are on the order of approximately $1 \times 10^{-5}$ duplications per cell division [14,42], suggesting that they are unlikely to have occurred in even one of the 200-generation *msh3Δ* MA strains assayed here. Only two of the haploid 2,000-generation MA strains we studied here had aneuploidies. Therefore, the most likely explanation for the high number of significant changes in growth rate observed among these strains is that they are the result of fitness consequences of SSR mutations. In combination with documented effects of SSR mutations and variants on gene expression in yeast [29] and humans [6,25–28], and the finding that SSR length is selectively constrained in laboratory populations of *Daphnia* [30], our study underscores the importance of looking beyond genic substitutions in studies aimed at understanding the phenotypic effects of mutations. Although not addressed in this study and although they make up a much smaller fraction of the genome, other repetitive sequences,

including those associated with telomeres and transposable elements, may also have higher frequencies of mutations and thus contribute to the distribution of fitness effects observed in mutation accumulation strains.

An important question addressed by our study concerns the frequency of mutations affecting complex traits and fitness (these traits' genomic target size). If a large proportion of mutations alter a trait value, it suggests that cellular networks are highly interconnected, because changes to genes in many pathways affect the trait [4]. Fitness is rightly seen as the ultimate interconnected trait [49], and the number of mutations affecting growth rate likely serves as a close lower bound of the number affecting fitness itself. However, estimates of the target size for fitness differ substantially. For example, Lynch and colleagues [50] estimate that 0.1% to 1.8% of all mutations have "discernible" fitness effects in *S. cerevisiae*. This low estimate, however, is at odds with our results. Although the overall rate of substitutions in the MA strains assayed here was independently estimated from sequencing data, the number of those mutations that had an effect on the phenotype in question was fit as a parameter in our model: the proportion of SNMs that are neutral ($s = 0$). This parameter can be interpreted in terms of the genomic target size for our phenotype of interest: mutations in ($1 - p_0$) of the genome result in effects on growth rate in rich media. Our maximum-likelihood estimate of $p_0$ is 6%, which implies that the majority of SNMs are non-neutral. In addition, estimates based on the maximum-likelihood parameter values of the DME suggest that >40% of SNMs would be visible to selection in the wild, where yeast population sizes are on the order of $3 \times 10^6$ [47]; the effective population sizes are even larger in typical laboratory evolution experiments, in which $N_e$ of >$10^8$ are common (e.g., [51,52]).

The uncertainty in the estimate of the genomic target size is important to consider with respect to the shape of the DME. Recent studies of mutational effects in *C. reinhardtii* and *E. coli* that also utilized mutation-number information found, as we did, that the DME is skewed toward very small effects [17,21]. It is therefore important to determine whether these very small effects are real or are based on faulty inference from noisy data. Unlike other parameters we estimated, the estimate of $p_0$ has a very large uncertainty that allows both possibilities: although the maximum-likelihood estimate of 6% suggests that very few SNMs are completely neutral, its 95% confidence interval ranges from 0% to 94% of SNMs. Any amount of measurement noise makes it difficult for a model to differentiate very small-effect mutations from zero-effect mutations. Nonetheless, other considerations lead to the conclusion that neutrality is limited. Despite the uncertainty in the proportion neutrality among individual mutations, we find that a large proportion of MA strains (approximately 60%) deviate significantly from their ancestor in growth rate, although most contain only approximately 4 SNMs, providing evidence that non-neutral mutations must be relatively frequent. Furthermore, our work suggests that a large number of mutations in SSRs have a significant effect on fitness: despite the fact that the average SSR mutation number in the *msh3Δ* MA strains assayed here is approximately 1.5 (with approximately 20% of strains expected to have no mutations at all), approximately 60% of these strains have detectable changes in growth rate relative to their ancestor. Overall, our results indicate that mutations affecting fitness are very common, both among SNMs and additional unidentified mutations.

We also demonstrate that the vast majority (73% to 99%) of non-neutral spontaneous substitutions affecting growth rate in rich media are deleterious. This result is consistent with most previous studies of the distribution of mutational effects on fitness phenotypes [9,17,31,53]. In our experiments, growth rate was assayed in rich media in which laboratory yeast are commonly propagated. The high proportion of deleterious mutations suggests that the strain used is already well-adapted to these growth conditions.

A key goal of mutation accumulation studies is that the distribution of mutational effects gleaned from these experiments would shed light on constraints affecting the speed and

direction of evolution. The low frequency of mutations increasing fitness identified in such experiments—including the present one—often precludes a detailed analysis of the properties of mutations that may be selected for in the course of evolution. Nonetheless, the distribution of mutations with deleterious effects on fitness is informative. During clonal evolution—long bouts of which occur in many microorganisms, as well as during tumor development—deleterious mutations have a direct effect on evolutionary trajectories due to the prevalence of mutational hitchhiking. When beneficial mutations are rare, they often appear in individuals harboring a number of deleterious mutations. As a result, the "fittest" individuals after a bout of clonal evolution often contain numerous deleterious mutations. Studies of cancer evolution show that these hitchhikers can have a significant effect on evolutionary trajectories, and may explain phenomena such as spontaneous cancer remission and tumor heterogeneity [8,54].

Our study can help untangle conflicting evidence regarding the prevalence of deleterious-mutation hitchhiking. A study comparing sexual and asexual evolution in yeast found that evolved asexual lines had significantly lower fitness than sexual lines, an effect attributed to hitchhiking deleterious mutations [55]. In contrast, a study that exhaustively measured the effects of hitchhiking mutations in a set of yeast lines evolved asexually found that only approximately 3% of these had significantly deleterious effects, with only a single mutation of 116 assayed decreasing fitness by >1%, while approximately 80% were neutral [56]. Our work can bridge the gap between these results, because it underscores the fact that most mutations are likely deleterious but with very small effects that are below the detection limit of commonly used growth assays. The DME estimated here implies that only 7.5% of all mutations can be expected to have individual effects with an $s < -0.01$ (1% decrease in growth rate), and about half of these would have deleterious effects strong enough to counteract the typical beneficial mutation identified by Buskirk and colleagues [56], making them targets of selection. However, we would expect another 88% of mutations to have deleterious effects smaller than 0.01 (similar to the 80% reported as neutral), and although the effects of these mutations would not be individually detectable in most assays, their combined effects would significantly decrease the fitnesses of the clones carrying them, consistent with the findings by McDonald and colleagues [55]. Thus, robust estimates of the distribution of mutational effects can provide key parameters for interpreting and potentially predicting the results of evolutionary processes.

## Materials and methods

### Strains

All strains used in this study are derived from MA lines described in [31,41] and [38]. Briefly, the original lines were created from a haploid strain of genotype *ade2 lys2-801 his3-Δ200 leu2–3.112 ura3–52 ho* that was transformed with a plasmid expressing HO to create a diploid strain homozygous at every locus genome-wide, with the exception of the mating type locus [31]. This strain was then passaged independently in 151 replicates by streaking a single colony every 2 days for approximately 2,000 generations on YPD media ("2,000-generation" experiment) [41]. To generate MA lines deficient in SSR slippage repair, the *MSH3* gene was deleted in a strain (*MAT0.a1*) derived from a spore of the diploid ancestor of the 2,000-generation experiment, and the resulting strain was passaged independently in 36 replicates as above for approximately 200 generations [38]. Because respiring *ade2* mutant colonies are red, petites could be detected by their white color and were not passaged to the next generation. To avoid unconscious bias in the passaging procedure, the red colony closest to a pre-marked spot on the plate was chosen at every passage [31,38].

The haploid 2,000-generation *MAH* strains used in this study were created previously: a subset of the MA lines were sporulated, and a random spore of *a* mating type was selected

[40,57]. Two *a* mating type spores from a full tetrad derived from the diploid ancestor, *MAT0. a1* and *MAT0.a2*, were phenotyped alongside the 2,000-generation *MAH* lines. These 2 strains were also streaked out onto YPD plates, and 6 independent white colonies on these plates were picked and used as petite control strains in the 2,000-generation growth assays; for the 200-generation assays, 3 petite colonies from the *MAT0.a1* strain were selected for use as the microcolony phenotyping assay petite controls.

To construct a GFP-marked ancestral reference strain, a *GFP* gene driven by an *Scw11* promoter was cloned upstream of a *HERP1.1* positive/negative selection cassette [58] that was flanked by 2 *Cyc1* terminator sequences. This construct was inserted into a neutral locus (*YFR054C* dORF) [34] in either the diploid ancestor or the *msh3Δ* haploid ancestor for the 2,000 and *msh3Δ* MA experiments, respectively, and, after selection on hygromycin and genotyping, yeast were re-selected on 50 μg/ml 5-fluorodeoxyuridine. The resulting strain was *YFR054C/YFR054CΔ::pScw11-GFP-Cyc1T* in the ancestral background of the 2,000-generation MA experiment, or *YFR054CΔ::pScw11-GFP-Cyc1T msh3Δ* in the ancestral background of the *msh3Δ* MA experiment. For the 2,000-generation experiment, this strain was sporulated, and progeny of a haploid, GFP-marked, *a*-mating type spore was selected to act as the in-well ancestral reference strain control.

For the *msh3Δ* MA experiment, we previously reported that we have likely identified every substitution in these strains that falls into a region of the genome that was sequenced at 10× or higher and that is not repetitive (telomere, centromere, or long terminal repeat) [38]. We therefore selected 28 "high-coverage" strains in which no more than 10% of the genome was sequenced at <10× or was part of a repetitive sequence, and then selected strains with no further mutations either at the final (200th) generation of MA or at the 100th MA generation.

## Growth rate assays

Strains were randomized into 96-well U-bottom plates and stored frozen at –80˚C in 20 μl of 50% YPD + 15% glycerol. "Petite-only" control strains were included in each experimental plate. Three days before each growth rate assay, a plate each of MA and reference strains was thawed and 180 μl SC media supplemented with 50 mg/l adenine (SC+Ade) added to each well (adding adenine decreases selection pressure for Ade+ phenotypes, including [PSI+] cells). After 1 day of growth in a shaking incubator at 30˚C, each strain was diluted 1:10 in SC+Ade in a new plate. The experiment was performed following an additional 2 days of growth from the resulting saturated cells of each line.

The microscope growth rate assay was performed largely as described in [35]. On the day of the growth rate assay, MA line and reference strains were mixed in a 2:1 ratio and diluted approximately $1 \times 10^{-4}$-fold with vigorous mixing. For the 2,000-generation experiments, in an attempt to identify petite microcolonies directly rather than through statistical modeling, strains were stained with MitoTracker Red CMXRos dye for 10 min as described in [33], followed by $10^{-4}$ dilution in SC media. However, this treatment was found not to efficiently stain mitochondria in the high-throughput experimental setup used here, so the fluorescence values were not used and the staining was not repeated for the *msh3Δ* MA experiment. The Mito-Tracker dye does not affect microcolony growth [33]. Cells were imaged hourly for 10 h in brightfield, followed by a single GFP exposure, as described previously [34]. Image analysis was performed using the PIE software [39]. *PIE* settings are listed in **S1 File**.

## Petite proportion experiments

To independently estimate the proportion of petites in 200-generation *msh3Δ* MA strains, we used a subset of the cells cultured on 2 experimental days and plated approximately 200

colonies per plate onto YPD plates. We tested at least 3 replicates of each strain per experimental day. Plates were grown for 2 days at 30°C and then left at room temperature for an additional 8 days before counting. Red colonies were counted as wild type, whereas white colonies were counted as petites.

To estimate the significance of variance in non-red colony proportion across strains as measured by the colony color assay, we fit a mixed effect logistic regression to estimate the odds of a colony being non-red, with strain as a random effect. We compared this model to one without the strain effect by a likelihood ratio test.

To estimate the significance of the improved fit of the individual line effect model when accounting for the existence of petites in the 2,000-generation MA experiment, we compared likelihoods of a "full" model, in which MA strain growth rates were estimated as described in **Eq 3** below, to a model in which all petite proportions were set to 0, meaning that MA strain growth rates were estimated as a Gaussian with a common standard deviation across all MA strains and a strain-specific mean.

### Sequencing and mutation calling

MA strains were grown for 2 days in YPD media, treated for 1 h in 600 µl sorbitol solution (0.9 M sorbitol, 0.1 M Tris (pH 8.0), 0.1 M EDTA) with 75 µl 100 T zymolyase and 1 µl β-mercaptoethanol, and yeast DNA was isolated using a Qiagen DNeasy Blood and Tissue kit (#69506). DNA sequencing and mutation calling was performed largely as described in [38]; briefly, DNA libraries were prepared following the protocol in [59], but with 14 rounds of PCR amplification. Bead cleanup optimized for 500 to 600 bp fragments was used, as described in [38]. Sequencing was performed on an Illumina NovaSeq 6000 SP flowcell with paired-end, 150-bp reads. Two strains (MAH.138 and MAH.150) were removed from further analysis due to low sequencing coverage.

To call mutations, we followed the mutation-calling pipeline described in [38], but using the S288C sacCer3 genome assembly as a reference. Briefly, mutations were called as differences relative to the ancestral-strain derived *MAT0.a1* sequence. Non-repeat mutations identified in multiple strains were removed, and SSR loci in the lower 35% percentile of call confidence levels were also removed from further analysis. Because our initial SSR analysis only included SSRs with motif length of 4 or smaller, indels in SSRs with larger or more complex motifs were sometimes called as part of our non-repeat mutation calling pipeline. Such indels were included with the list of SSR mutations for the purposes of the predicted mutational impact analysis described above.

To detect aneuploidies, the coverage in 10-kb windows across the genome was computed using bedtools [60]. The $\log_2$(coverage) of each window (relative to the average coverage across all strains) was then modeled as a function of the window identity and strain. Model residuals were averaged across all the windows on a chromosome, and chromosomes in a strain having an absolute value of the $\log_2$(relative coverage) mean residual value above 0.3 were further inspected. In practice, all aneuploid chromosomes in our dataset had a $\log_2$(relative coverage) mean residual value close to 1, indicating duplication of the entire chromosome.

To identify putative mutational effects, we ran snpEff [44] on the mutations identified in our data. snpEff identifies multiple possible putative mutational effects for each mutation, taking into account different nearby genes. For each mutation, we selected the most severe effect identified and its corresponding gene (this was always the nearest gene to the mutation as well). When assessing the effects of multiple mutations in a gene, or multiple mutant genes in a strain, we also identified the most severe of the putative effects listed: e.g., a gene with 1 early stop codon ("high" impact) and 1 nonsynonymous substitution ("moderate" impact) in a particular strain was labeled as having a high-impact mutation.

Analysis code and data necessary to replicate these findings can be found on the Open Science Framework: https://doi.org/10.17605/OSF.IO/H4J9F. All sequencing data was deposited in the Sequence Read Archive under project PRJNA1117962.

## Outline of parameter estimation

We are interested in identifying the values of the parameters of the DME, $\theta_{DME}$, based on a set of observations of colony growth (here generally denoted $O$). Within a single strain $i$, this probability is the convolution of the probability of mutational effect $S_i$ (e.g., as defined in **Eq 1**) being drawn from a DME with parameters $\theta_{DME}$, and the probability of the set of observations of strain $i$ given $s$:

$$P(O_i|\theta_{DME}) = P(S_i = s|\theta_{DME}) * P(O_i|s), \qquad (2)$$

where * denotes convolution. The overall likelihood of $\theta_{DME}$ is the product of all the likelihoods over strains $i$, and is maximized by maximizing the sum of the log likelihoods across all strains, as discussed in detail below.

Below, we first outline the computation of the probability of colony growth observations given $s$. We then describe the computation of the distribution of mutational effects for different classes of candidate probability functions. We present 2 ways in which we combined these 2 functions (one summary statistic-based, and one jointly performing the likelihood computation across every individual observation) to calculate the total likelihood of a set of $\theta_{DME}$ values given a set of growth observations. Finally, we briefly describe the computational methods used to perform likelihood maximization. Note that parameters in $\theta_{DME}$ are described in **S3 Table**, and other parameter names used in this section are listed in **S7 Table**.

## Computation of the likelihood of single-strain growth rate observations

In this section, we describe the calculation of the probability of making growth observations of a strain given its mutational effect, $P(O_i|s)$. We discuss the need to account for the fact that growth rate distributions consist of populations of both respiration-deficient (petite) and respiring colonies, as well as for the substantial batch effects in growth rate estimation. We show that using the differences in growth rates between randomly paired MA and reference strain colonies as the observations of interest allows us to account for both factors in a computationally efficient manner.

**Accounting for petites.** The colonies of a single strain consist of 2 distinct subpopulations that grow at different rates. The first subpopulation, constituting the majority of the population, consists of cells with a strain-characteristic growth rate for each strain $i$ ($\mu_i$) and with a standard deviation that we model with a parameter that is common to all strains within an MA experiment ($\sigma_{nonpetite\text{-}bio}$). Each population also contains a smaller proportion of petite colonies, which have lost the ability to respire and grow at a lower rate than their non-petite counterparts in fermentation conditions. We model these petite cells as having a common mean growth rate ($\mu_{petite}$) and standard deviation ($\sigma_{petite\text{-}bio}$) across all strains. Experimental data show that the strains in our experiment vary in the proportion of petites, $\rho_i$. Therefore, we model the growth rate, $g$, of each colony $a$ of strain $i$ as being drawn from the distribution:

$$g_{ai} \sim (1 - \rho_i)N(\mu_i, \sigma^2_{nonpetite-bio}) + \rho_i N(\mu_{petite}, \sigma^2_{petite-bio}). \qquad (3)$$

The setup of the growth rate assays described in this paper is such that $\rho_i$ may vary among strains for nonbiological reasons: After the completion of the mutation accumulation, the strains are passed through a relatively narrow bottleneck before being frozen, followed by 6 to 8 generations of subculture. As a result, $\rho_i$ may vary as a result of jackpotting events rather

than underlying biology. Because of this, we chose to focus on understanding mutational effects affecting $\mu_i$, not $\rho_i$, such that **Eq 1** can be restated as follows:

$$s_i = \frac{\mu_i}{\mu_{anc}} - 1. \tag{4}$$

Note that although $\rho_i$ is not the focus of our study, it is important to estimate this strain-specific proportion of petites in order to accurately estimate $\mu_i$, and it is therefore estimated for each strain as part of our model fitting.

**Accounting for batch effects and colony measurement noise in experimental setup.** The growth rate assay is performed in batches. In the assay for the 2,000-generation experiment, each MA strain is assayed in a single well on an experimental plate, with a total of 14 experimental plates assayed. In the assay for the *msh3Δ* experiment, each MA strain is assayed in 3 wells per plate, and the ancestral strain is present in 9 wells per plate, with a total of 10 experimental plates assayed. For each well, growth rates were typically obtained for approximately 500 microcolonies of the strain being assayed as well as approximately 500 microcolonies of a single GFP-marked reference strain.

Previous work has shown that the experimental organization can result in significant batch effects across wells within a plate and across plates (which correspond to experimental days) [36]. These batch effects can be accounted for in the context of a mixed effect model:

$$g^{obs} \sim N(\overrightarrow{\mu}, \Sigma) \tag{5}$$

Where $g^{obs}$ is the vector of measured (observed) growth rates, $\overrightarrow{\mu}$ is the vector of mean growth rates, and $\Sigma$ is the covariance matrix of colony growth rates derived from the block design.

Colonies from "petite-only" samples can be analyzed in the above way. However, as described in **Eq 3**, the growth rates of MA and reference strains are described by bimodal distributions with an unknown proportion of petites, complicating the computation of the likelihoods of observing specific growth rates.

If random batch effects are additive on the growth rate scale, then their effects can be eliminated by calculating the difference in growth rate between a reference strain colony and an MA colony that share the same experimental plate and well. For example, consider the observed growth rates, $g_{ai}^{obs}$ and $g_{bj}^{obs}$, of 2 colonies $a$ and $b$ of MA strain $i$ and reference strain $j$, respectively, growing in the same microscope plate $k$, in well $l$. The genetic component of the growth rate of each colony is distributed as described in **Eq 3**: 2 sources of measurement error are responsible for the difference between observed and "genetic" growth rates: $\varepsilon_{kl}$, which is the sum of the batch effect of the plate and well in which the colonies are grown (with standard deviations $\sigma_{plate}$ and $\sigma_{well}$, respectively) and an independent and identically distributed (iid) measurement noise across colonies (with standard deviation $\sigma_{col}$):

$$g_{ai}^{obs} = g_{ai} + \varepsilon_a + \varepsilon_{kl}$$

$$g_{bj}^{obs} = g_{bj} + \varepsilon_b + \varepsilon_{kl}$$

$$\varepsilon_{a,b} \sim N(0, \sigma_{col}^2)$$

$$\varepsilon_{kl} \sim N(0, \sigma_{plate}^2) + N(0, \sigma_{well}^2), \tag{6}$$

(note that indices $kl$ were left off of $g_a$ and $g_b$, respectively, to streamline notation).

Because the colony measurement terms $\varepsilon_a$ and $\varepsilon_b$ are iid random variables, we can subsume them into a combined noise term that includes both the biological variation in colony growth rates and the independent cross-colony measurement noise:

$$\sigma^2_{nonpetite} = \sigma^2_{col} + \sigma^2_{nonpetite-bio}$$

$$\sigma^2_{petite} = \sigma^2_{col} + \sigma^2_{petite-bio}. \tag{7}$$

This eliminates the $\varepsilon_a$ and $\varepsilon_b$ terms from **Eq 6**, although it also means that we are not able to independently estimate the biological and measurement noise components of petite and nonpetite colony growth variance.

Next, subtracting the growth rates of colonies $a$ and $b$ from each other yields the formula for the growth rate difference, $D$, eliminating the random batch effect term $\varepsilon_{kl}$. Thus, by estimating the likelihood of the difference in growth rates between 2 colonies that share random batch effects, $D$, we eliminate the necessity of using a mixed effect model to estimate likelihood. Instead, the probability of observing a particular difference in colony growth rates between a same-well colony pair becomes dependent on a mixture of independent normal distributions:

$$D \sim (1 - \rho_i)(1 - \rho_j)N(\mu_i - \mu_j, 2\sigma^2_{nonpetite})$$

$$+ (1 - \rho_i)(\rho_j)N(\mu_i - \mu_{petite}, \sigma^2_{nonpetite} + \sigma^2_{petite})$$

$$+ (\rho_i)(1 - \rho_j)N(\mu_{petite} - \mu_j, \sigma^2_{nonpetite} + \sigma^2_{petite})$$

$$+ \rho_i\rho_j N(0, 2\sigma^2_{petite}). \tag{8}$$

We select random pairs of colonies without replacement from an MA strain and the GFP-marked reference strain within each imaging field in each well and estimate the likelihood of the parameter values above given each pair's difference in growth rates.

The logs of these likelihoods are then added to the sum of the log likelihoods of the petite parameters given each colony growth rate in the petite-only samples to calculate the total likelihood of our data. To identify the maximum likelihood parameter values, we iterate over values of each of the following "general" parameters: $\mu_{ancestor}$, $\mu_{petite}$, $\rho_{ancestor}$, $\sigma_{nonpetite}$, $\sigma_{petite}$, $\sigma_{plate}$, $\sigma_{well}$; and find the maximum likelihood parameter values of the MA strain petite proportions $\rho_{MA}$ and selection coefficient $s_{MA}$ for each MA strain (as defined in **Eq 4**). We then find the values of the "general" parameters that result in the maximum overall likelihood.

In the 2,000-generation experiment, the GFP-marked reference strain was treated as the "ancestral" strain, and indeed this strain did not differ significantly in growth rate from 2 haploid strains derived from the diploid MA ancestor. In the 200-generation *msh3Δ* experiment, the in-well GFP reference strain was found to have a different growth rate than the ancestral *msh3Δ* strain, so the *s* value for each MA strain was estimated relative to the ancestral *msh3Δ* strain.

## Computation of the probability density function for the distributions of mutational effects

To jointly estimate the distributions of effect sizes of SNMs and unidentified mutations based on the MA lines, we expanded on a modeling approach developed by Keightley [11,12]. As described in Results, we model the distribution of mutational effects as a reflected gamma

distribution with shape $k$ and mean $m$; the 2 sides of the distribution are weighted by the proportion of non-neutral beneficial mutations $q$. The distribution of MA line phenotypes is the convolution of a Poisson random number of single DME distributions with mean $U$.

Previous work has found that fitting the model described in [12] is nontrivial due to the significant amount of time required for the numerical integration used to estimate the combined probability density function (PDF) of mutational effects caused by multiple mutations. Rather than performing multiple numerical integrations, we transferred the computation of the density function into the Fourier domain, as follows.

First, consider a mutation whose mutational effect, $Y$, is gamma-distributed. We can describe the PDF of observing a mutational effect $s$ of this mutation using its characteristic function (the Fourier transform of the PDF):

$$F_Y(\omega) = \frac{1}{\left(1 - i\omega \frac{m}{k}\right)^k} \tag{9}$$

where $m$ and $k$ are the mean and shape parameters of the distribution, as described in the Results section.

We would like to instead model the distribution of single mutational effects $Z$ as a reflected gamma distribution weighted by $q$ and $1$-$q$ for positive and negative mutational effects. The characteristic function of $Z$ is then

$$F_Z(\omega) = \frac{1-q}{\left(1 + i\omega \frac{m}{k}\right)^k} + \frac{q}{\left(1 - i\omega \frac{m}{k}\right)^k} \tag{10}$$

For a strain with a known number of mutations, $n$, the combined mutational effect $S$ can be expressed as follows:

$$S = Z_1 + Z_2 + \cdots + Z_n.$$

The PDF of $S$, $f_S$, is the convolution of $n$ PDFs of $Z$, and the characteristic function of $S$ in this known-mutation number case is thus

$$F_S^{known\ mut}(\omega) = (F_Z(\omega))^n. \tag{11}$$

The mean number of mutations across the parental diploid strains has been precisely estimated [42]. We therefore treat the number of non-neutral mutations in each MA strain as a Poisson-distributed random variable, $N$, with mean $U$:

$$f_N(N = n) = \frac{U^n}{n!} e^{-U}.$$

For a set of strains each containing an unknown number of non-neutral mutations drawn from a single Poisson distribution with mean $U$ the characteristic function of the combined mutational effect $S$ observed in the MA strains is therefore

$$F_S^{SNM-only}(\omega) = F_S^{single\ DME}(\omega) = \sum_{n=0}^{\infty} \frac{(UF_Z(\omega))^n}{n!} e^{-U} = e^{UF_Z(\omega) - U} = e^{U(F_Z(\omega) - 1)} \tag{12}$$

This equation applies to both the "*single DME*" model, in which $U$ is a parameter estimated by the model, and the "*SNM-only*" model, in which $U$ is a function of $M_{diploid}$ (the mean number of mutations per diploid MA strain) and the parameter $p_0$ (the probability that any single

mutation is neutral, with $s = 0$), which is in turn estimated by the model:

$$U^{SNM-only} = \frac{M_{diploid}}{2}(1 - p_{0\ SNM}).$$ (13)

For the models that treat SNMs and unidentified mutations as being drawn from distinct distributions, $f_S$ is a convolution of the distributions of SNMs and unidentified mutational effects in each strain; thus, for the "*two-gamma*" model described in the text, there are 2 distinct reflected gamma distributions with different parameters, one for the sequenced SNMs ("*SNM*") and one for the unidentified mutations ("*unid*"):

$$F_S^{two-gamma}(\omega) = e^{U^{SNM}(F_Z^{SNM}(\omega)-1)}e^{U^{unid}(F_Z^{unid}(\omega)-1)},$$ (14)

and for the "Gaussian" model described in the text, the unidentified mutational effects are normally distributed:

$$F_S^{Gaussian}(\omega) = e^{U^{SNM}(F_Z^{SNM}(\omega)-1)}F_N^{unid}(\omega).$$ (15)

Where $F_N^{unid}(\omega)$ is the Fourier transform of a Gaussian PDF with mean and standard deviations $\mu_{unid}$ and $\sigma_{unid}$, respectively.

## Estimation of the distribution of mutational effects from MA-line-wise summary statistics

One approach described in the Results section for calculating the likelihood of the observed growth data was to first estimate a mutational effect $S_i$ and standard error of the $S_i$ estimate for each 2,000-generation MA strain, and then use sampling distributions parameterized by these values as the probability of the observed growth rate differences in **Eq 2**. To calculate the likelihood of the observed results, the distribution of MA line phenotypes (relative to the ancestral strain) is convolved with a normal distribution whose standard deviation is the error estimate based on the confidence intervals of each line's mutational effect $S_i$, following [11,12], here with an individual strain-specific error estimate derived from likelihood profiles of each strain's mutational effect estimates. In practice, $f_S$ is also convolved with a narrow Gaussian kernel to resolve numerical issues that occur in computing $f_S$ in certain parts of parameter space.

As a result, the characteristic function of the estimated mutational effect of MA strain $i$, $S_i^{est}$, is

$$F_{S_i^{est}}(\omega) = F_S(\omega)F_N^{kernel}(\omega|0, 2^{-16})F_N^{error}(\omega|0, \sigma_{MA_i}).$$ (16)

Where $F_S(\omega)$ is given by **Eqs 12**, **14**, or **15** above, depending on the model.

By computing this convolution in the Fourier domain, and then performing a discrete inverse Fourier transform, we compute the PDF of estimating a mutational effect given a set of DME parameter values. We then interpolate estimated mutational effects measured in each MA strain, $S_i^{est}$, (**S1 Table**) within the computed values for this density function to compute the likelihood of each mutational effect estimate. This likelihood is maximized across all parameters shown in **S3 Table**.

## Estimation of the distribution of mutational effects from complete colony growth data

To directly jointly estimate the likelihood of observing $v$ differences $D^{(i)}_{1\ldots v}$ in growth rates between each MA strain colony of a single strain $i$ and ancestral reference colony given a

distribution of mutational effects $f_S$, we computed the likelihood of the DME parameters ($\theta_{DME}$), the strain-specific petite proportions, and the "general" parameters described above ($\theta_{general}$, which includes the mean petite and ancestral strain growth rates, the petite proportion of the reference strain, the standard deviations of petite and non-petite colony growth rates, and random batch effect standard deviations), given the observed set of growth rate differences between random pairs of MA and reference colonies in each well:

$$P(observed\ growth\ differences = D_{1...v_i}^{(i)}|\theta_{DME}, \theta_{general}, \rho_i)$$

$$= \int P(mutational\ effect\ S_i = s|\theta_{DME}) * (\prod_{j=1}^{v_i}[P(observed\ growth\ difference$$

$$= D_j^{(i)}|\theta_{general}, s, \rho_i)])ds \tag{17}$$

where $D_j^{(i)}$ is the $j$<sup>th</sup> difference observed in strain $i$, and $v_i$ is the number of observed differences (colony pairs) in strain $i$.

The integral above was computed using numerical integration, with the individual probabilities inside computed as described above: using **Eqs 12**, **14**, or **15** to calculate the PDF over $s$ for the first half of the integral and **Eq 8** to calculate the probability inside the product. Note that the random variable $\mu_i$ from **Eq 8** appears in every term of the product in **Eq 17** and that the change of variable

$$\mu_i = \mu_{ancestor} * (1 + S_i)$$

is done before doing the integration in **Eq 17**, also within the product. All of the terms within the integral are initially calculated as log likelihoods and only exponentiated for integration. The log likelihood of the complete set of measurements across strains is computed by summing over all strains the log of the likelihoods as in **Eq 17**; for each term, the strain-specific set of $D$ values is used.

The maximum likelihood parameters were identified by iterating over sets of parameter values in a nested algorithm similar to that described above for strain mutational effects: the maximum likelihood parameter values of $\rho_{MA}$ are identified for fixed values of $\theta_{DME}$ and $\theta_{general}$; the maximum likelihood of $\theta_{DME}$ are found for a fixed set of values of $\theta_{general}$.

## Maximum likelihood parameter value and confidence interval identification

Maximum likelihood parameter values were identified by implementing an interior-point minimization algorithm in MATLAB and minimizing the negative log likelihood [61], using constrained optimization with wide bounds on possible parameter values. Optimization code can be found at https://github.com/plavskin/MutationEffectEstimation [62] and likelihood calculation code can be found at https://github.com/plavskin/GR_diff_DFE [63]. The data and additional analysis code can be found on OSF: https://doi.org/10.17605/OSF.IO/H4J9F.

Confidence intervals on all parameter values were calculated using the profile likelihood method following [11], where the value of a single parameter is fixed at various points, and likelihood is maximized with respect to all other parameters, and 95% confidence interval bounds were then identified via quadratic interpolation between points along the parameter of interest to identify the point corresponding to a log likelihood change of approximately 2.5.

To ensure global likelihood maxima were identified, constrained maximum likelihood estimation was started at multiple points across most parameters. The maximum likelihood parameter search was repeated multiple times, starting at the optimal parameter values from

the previous run and additional points in parameter space, until a consistent maximum likelihood value was identified regardless of starting point, and each parameter's log likelihood profiles were monotonically ascending from the left and monotonically descending to the right of the maximum likelihood parameter values.

## Supporting information

**S1 File. S1_file_S1_file_PIE_setup_file.csv.** *PIE* setup file used for image analysis.
(CSV)

**S2 File. YFR054C-Scw11p-GFP-Cyc1T-HERP1.1_delitto_perfetto.str.** Sequence of *YFR054C* locus after insertion of full GFP marker cassette and hygromycin selection.
(STR)

**S3 File. S3_file_YFR054C-Scw11p-GFP-Cyc1T-post-dp.str.** Sequence of *YFR054C* locus with GFP marker cassette after FuDR counterselection and resistance cassette removal.
(STR)

**S1 Table. Estimated mutational effect of each 2,000-generation mutation accumulation strain.** The mutational effect *s* and petite proportion for the single haploid 2,000-generation MA progeny phenotyped in this study (MAH lines), as well as 2 haploid lines derived from the 2,000-generation mutation accumulation ancestor (MAT.0.a1-a2); 95% confidence interval presented in parentheses. Two strains whose sequencing data was not analyzed due to poor quality are missing ploidy information.
(XLSX)

**S2 Table. Non-repeat mutations in 2,000-generation mutation accumulation strains and their putative effects predicted by snpEff.** When multiple mutations are within 50 bp of each other, they share a "superlocus." The snpEff annotation with the most severe putative effect is listed for each mutation; see [44] for details on additional snpEff output columns.
(XLSX)

**S3 Table. Properties of DMEs identified by various models using mutational effect summary data.** Preliminary models fitted to summary data of individual strain *s* value fits. Models that attempt to fit unidentified mutations either as additional mutations drawn from the same reflected gamma distribution as SNMs, or as an independent reflected gamma distribution to individual unidentified mutations, improve fit only marginally. The "*two-gamma*" model produces uninformative parameter estimates, with poor computational likelihood estimation across the likelihood profile (due to confounded parameters). The resulting non-monotonic likelihoods lead to uninterpretable confidence interval bound estimation for many parameters (question marks). The "Gaussian" model, which accounts for unidentified mutations as a Gaussian distribution representing the effects of these mutations across strains, performs better than a model that only accounts for SNMs. Parameter values for each model shown with 95% confidence intervals; ΔAIC is calculated relative to the "SNMs only" model.
(XLSX)

**S4 Table. Estimated mutational effect of each *msh3Δ* mutation accumulation strain.** The mutational effect *s* and petite proportion for *msh3Δ* MA strains phenotyped in this study, as well as the *MSH3⁺* strain from which the *msh3Δ* ancestor was derived (a clone of *MAT0.a1*), and a GFP-marked *msh3Δ* strain used as an in-well reference; 95% confidence interval presented in parentheses and % genome sequenced at >10× and not part of a repetitive sequence

comes from data in [38].
(XLSX)

**S5 Table. SSR repeat mutations in *msh3Δ* mutation accumulation strains and their putative effects predicted by snpEff.** Indels in SSR loci with repeat size >4 nucleotides, or in complex repeats, are included in this table but do not have an associated SSR start and end listed. The snpEff annotation with the most severe putative effect is listed for each mutation; see [44] for details on additional snpEff output columns.
(XLSX)

**S6 Table. SSR repeat mutations in 2,000-generation mutation accumulation strains and their putative effects predicted by snpEff.** Indels in SSR loci with repeat size >4 nucleotides, or in complex repeats, are included in this table but do not have an associated SSR start and end listed. The snpEff annotation with the most severe putative effect is listed for each mutation; see [44] for details on additional snpEff output columns.
(XLSX)

**S7 Table. Variables used in modeling.** Variables used in modeling the DME or individual strain *s* values.
(XLSX)

**S1 Fig.** (A) Petite proportion estimated for strains from [38] using plate-based colony color assay versus by modeling observed colony growth rates in the microscope assay as a mixture of Gaussians. Error lines represent 95% confidence intervals; dashed line shows a 1:1 correspondence. Each point represents estimates for a single strain on a single experimental day (with error bars based on replicates across microscope plate wells). Two data points in red are for a strain in which colony color and colony size were decoupled. (B) A histogram of growth rates of the ancestral reference strain in all wells in which it was co-cultured with 2,000-generation *MAH* strains. Red line shows the distribution estimated by the best-fit model of the distribution of growth rates for this strain in the full model (including a distribution of petites); blue line shows distribution estimated by the model in which the distribution of growth rates is not partitioned into petite and non-petite growth rates. Note that although distributions are shown overlaid on raw reference strain growth rate measurements, the models that produced the distribution parameters were based on differences between reference strain and MA strain growth rates (see Methods). (C) Mutational effects for strains from **Fig 1** estimated either by the full model of MA strain *s* effects described in the text, or by a model that does not include a petite population in any of the strains. Error lines represent 95% confidence intervals; dashed line shows a 1:1 correspondence. Two ancestral control strains included in the experiments (purple points) have mutational effects whose confidence intervals overlap with 0 when petites are accounted for, but not when they are ignored.
(TIFF)

**S2 Fig.** The absolute value of the MLE of the selection coefficient of each 2,000-generation MA strain, with strains grouped by the effect of the highest putative effect non-repeat mutation, as predicted by snpEff, found in each one. Points are colored yellow if their *s* value differs significantly from the ancestor at an FDR of 0.05.
(TIFF)

**S3 Fig.** The cumulative density function of the fit of each DME model to all individual MA strain mutational effects *s*. Inset: histogram of mutational effects with probability density functions of the models overlaid. To account for the effect of experimental noise on the estimates of *s*, the model density function is shown convolved with a Gaussian noise kernel with a

variance that is the mean of the error variances of each strain's mutational effect estimate. *SNMs + unid*, *Gaussian* model as in **Fig 2**.
(TIFF)

**S4 Fig.** The MLE of the absolute value of the selection coefficient of each *msh3Δ* MA strain, with strains grouped by the effect of the highest putative effect SSR mutation, as predicted by snpEff, found in each one. Points are colored yellow if their *s* value differs significantly from the ancestor at an FDR of 0.05.
(TIFF)

## Acknowledgments

This work was supported in part through the NYU IT High Performance Computing resources, services, and staff expertise; we especially thank Shenglong Wang for support setting up the likelihood estimation code on the NYU High Performance Cluster. We thank the NYU Genomics Core for help with sequencing. We are grateful to Federica Sartori and Cassandra Buzby for comments on this manuscript and to Dmitri Petrov for helpful discussions.

## Author Contributions

**Conceptualization:** Yevgeniy Plavskin, David W. Hall, Daniel Tranchina, Mark L. Siegal.

**Formal analysis:** Yevgeniy Plavskin.

**Funding acquisition:** David W. Hall, Roland F. Schwarz, Mark L. Siegal.

**Investigation:** Yevgeniy Plavskin, Maria Stella de Biase, Naomi Ziv, Libuše Janská, Yuan O. Zhu.

**Methodology:** Yevgeniy Plavskin, Daniel Tranchina, Mark L. Siegal.

**Project administration:** Mark L. Siegal.

**Resources:** David W. Hall.

**Software:** Yevgeniy Plavskin, Daniel Tranchina.

**Supervision:** Roland F. Schwarz, Daniel Tranchina, Mark L. Siegal.

**Visualization:** Yevgeniy Plavskin, Mark L. Siegal.

**Writing – original draft:** Yevgeniy Plavskin.

**Writing – review & editing:** Yevgeniy Plavskin, Maria Stella de Biase, Naomi Ziv, Libuše Janská, Yuan O. Zhu, David W. Hall, Roland F. Schwarz, Daniel Tranchina, Mark L. Siegal.

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
