## [Editor Report · Decision Letter 0]

11 Jul 2023

Dear Mark, 

Thank you for submitting your manuscript entitled "Spontaneous single-nucleotide substitutions and microsatellite mutations have distinct distributions of fitness effects" for consideration as a Research Article by PLOS Biology.

Your manuscript has now been evaluated by the PLOS Biology editorial staff, as well as by an academic editor with relevant expertise, and I'm writing to let you know that we would like to send your submission out for external peer review.

Once your full submission is complete, your paper will undergo a series of checks in preparation for peer review. After your manuscript has passed the checks it will be sent out for review. To provide the metadata for your submission, please Login to Editorial Manager (https://www.editorialmanager.com/pbiology) within two working days, i.e. by Jul 13 2023 11:59PM.

Kind regards,

Roli

Roland Roberts, PhD

Senior Editor

PLOS Biology

rroberts@plos.org

---

## [Decision Letter · Decision Letter 1]

7 Sep 2023

Dear Mark,

Many thanks for your patience while your manuscript "Spontaneous single-nucleotide substitutions and microsatellite mutations have distinct distributions of fitness effects" was peer-reviewed at PLOS Biology. It has now been evaluated by the PLOS Biology editors, an Academic Editor with relevant expertise, and by three independent reviewers. 

You'll see that Reviewer #1 is positive, but raises a number of concerns. S/he requests an explanation for (and further characterisation of) the ‘petite’ mutants, and analysis of the type and location of the mutations (both SNMs and SSRs). Reviewer #2 is also positive and simply has some presentational suggestions. Reviewer #3 is similarly very positive, but asks for a stats test of whether the DFEs for SNMs and SSRs differ from each other.

As you may know, we now invite reviewers to cross-comment on each other's reviews, and this was helpful in guiding the Academic Editor and me to an idea of which of reviewer #1's substantial requests should be required. I've pasted a lightly edited version of the Academic Editor's advice at the foot of this letter, so that you can see the rationale. Essentially the short answer is that we don't need the investigation of the petites, but I'm afraid that we would need the analysis of mutation type/location.

In light of the reviews, which you will find at the end of this email, we would like to invite you to revise the work to thoroughly address the reviewers' reports.

Given the extent of revision needed, we cannot make a decision about publication until we have seen the revised manuscript and your response to the reviewers' comments. Your revised manuscript is likely to be sent for further evaluation by all or a subset of the reviewers.

**IMPORTANT - SUBMITTING YOUR REVISION**

*Re-submission Checklist*

*Published Peer Review*

*PLOS Data Policy*

*Blot and Gel Data Policy*

Best wishes,

Roli

Roland Roberts, PhD

Senior Editor

PLOS Biology

rroberts@plos.org

REVIEWERS' COMMENTS:

Reviewer #1:

Plavskin et al. attempt to estimate the distribution of fitness effects of de novo mutations in Saccharomyces cerevisiae. This question is indeed important for understanding the dynamics of evolution, quantitative traits' architecture, genetic load of extant populations, cancer development, etc. The authors generally correctly explain shortcomings of former studies. I have not noticed in the text any generalization about the distribution(s) of mutational effects reported until now with yeast so that the authors could conclude to what extent their finds confirm or contradict those already known. I believe the main finding of the study (s mostly negative and small) is correct, I would be highly surprised if something much different was found. It does not mean that the present experiment simply repeats those done earlier. The strongest aspect of the study is the application of highly repeated measurements of microcolony growth rate which helps to get precise estimates. The design of the experiment, including the use of controls, appears correct. Computational analyses built on earlier ones but try to introduce new constraints and are carefully done. Still, there are several non-trivial issues that need to be explained before the paper is published.

1) "Variation in petite numbers across samples can arise from chance events that cause different numbers…" A single loss of the mitochondrial function is probably "random". But, some strains have repeatedly about 10% while others up to 30%, even if two different methods are applied, colony color and distribution fitting. Kind of stabilized randomness. The authors say:" these strains differed from each other by only ~2 mutations on average, large variation in the proportion of petites across these strains was not likely to be explained by genetic differences among the strains." First of all, mutations were few but different. If not the few known mutations then what caused this variation. Epigenetics? Hardly. Repeatable elements, plasmids, viruses? Don't know. My guess would be mutations in mtDNA of the highly inbred lines. Was mtDNA analyzed with comparable accuracy to the nuclear DNA? Is it just my curiosity? The authors should suggest some explanation, at stake is the question how representative are their strains for "normal" yeast.

2) The petite proportions are extremely high. Why apparently none strain was put onto non-fermentable medium to confirm these were indeed petites? Petites are really slow growers so they are quickly overgrown by respiring cells in normal strains (about 1% of petites). With 0.1-0.3 proportion, the petite mutation rate must have been up to 0.5. If so, there were no two types of colonies; "petites" versus "non-petites". Rather there was a rising fraction of pure "petites" (arriving after last respiring within colony mutated) against a declining fraction of petite-respiring mixed colonies of evolving composition (with selection favoring non-petites but mutation killing them). Can you model all this mess to extract TRUE non-petite s? Or perhaps admit that petites are part of every strain and calculate overall s for all the beasts forming it? (As the poor "batchers" do.)

3) "(Pearson correlation coefficient = 0.83) indicating that microcolony growth rate data can be used directly to partition growth rates of petite and respiring colonies." Squared becomes not that impressive. Errors introduced here may account for much of the variation in the s of respiring cells. So, think of the advice from the previous point.

4) "diploid parent strains that accumulated mutations over the course of 2000 generations accumulating an average of ~8 SNMs." First, were these mutations in coding regions, non-synonymous and with likely non-negligible protein effect? If not, why do the authors use these numbers in their models? Yes, there was a famous paper in Nature claiming that synonymous may have handsome s, but I prefer the old framework. Second, among the parental diploids the scores of mutations must have differed (4-12?), was there any correlation between these counts and average s of the two derived haploids?

5) "Consistent with our previous finding, the 'Gaussian' model, which modeled SNMs and unidentified mutations as having two independent DMEs, provided the best fit to the data". Can you think of an analysis of empirical data in which adding two more parameters to a model would not improve the fit? 

6) "These strains accumulated mutations over the course of 200 generations, and contain an average of 1.8 SSR mutations each, the majority of which are deletions of a single repeat unit." Again, where they were located, what effect could they have. In the case of SSR the question is even more important because here "17% of strains have a significant increase in growth rate, and 44% have a significant decrease in growth rate". How can I accept it without seeing the mutations. MSH3 is known to stabilize many repetitive sequences, especially telomeres, those regular and their remnants scattered over chromosomes. NGS tells nothing of mutations within them and about many types of possible chromosome-segment mutations in mismatch repair deficient strains. I do not believe in the whole story about SSR, including the title. Yes, I can not prove I am right and you are wrong.

7) "potential aneuploidies were excluded from our analysis", from diploids, but meiosis could introduce them again especially in these pretty awkward strains

8) "Two a mating type spores from a full tetrad derived from the diploid ancestor". It would be helpful to know about the frequency of full uniforms tetrads or 2:2 segregations with lethals or slow growers. There was a study based on that. It lacked fancy models but provided results parallel to those reported here, https://doi.org/10.1093/genetics/159.2.441.

Reviewer #2:

The authors address challenges in the field of evolutionary genetics and fitness by using a sensitive assay and modeling to determine the effects of mutations on growth in S. cerevisiae. Their rationale and findings are clearly communicated, and address their 3 stated goals of determining the frequency of deleterious mutations, what proportion of the genome that affects growth when mutated, and how different classes of mutations affect growth. Their findings that many mutations are deleterious, but with very small effect sizes, make an important contribution to reconcile prior studies that differed in their findings because many of these small effect sizes are not captured in other fitness assays. Their method and models are a powerful tool for investigation of growth phenotypes, particularly by addressing the contribution of petites. This study provides resources useful to the yeast genetics field and results with broad impact across the field of evolutionary biology.

My only minor comments, which do not necessitate revision but should be addressed if others deem revision or edits are needed, are to clarify figures labels. The labeling of Fig. 2B, in the panel and the written legend, was unclear - the label "without petites" seems to indicate that petites were excluded from the data used to generate the model, which is not the case. "Without partitioning," or "Single gaussian" and "Two gaussian" would make it clearer which line goes with which model as described in the main text. In general, adding legends showing color and label into the panels themselves would make reading and interpretation easier than searching for them in the written figure legends (e.g., how Fig 2B, S1 Fig B, and S2 Fig have the color legends within the panel).

Reviewer #3:

[identifies himself as David Houle]

This manuscript addresses the very interesting issue of the distribution of effects of spontaneous mutations on fitness. The strengths of this paper are many. 

It leverages the many tools available in the yeast system in service of the ultimate goal. This enables the authors to preserve and label control strains, to account for the annoying tendency of yeast strains to throw off petite genotypes, and to obtain wonderful precision in the estimates of growth rates of individual clones. Finally, by disabling the slippage repair mechanism, they accumulated mutations in repeat numbers at rates large enough to provide a useful sample. 

The authors are able to count single nucleotide mutations, and use those totals to put precise limits on their estimates of the DFE. 

The results provide very useful estimates of the proportion of SNMs with positive and negative effects. The result that the most likely proportion of neutral mutation is very low is quite striking. The broad confidence limits on this estimate are disappointing, but the result itself is quite provocative, and will surely lead to additional work.

Some of the most useful results are the authors estimates suggesting the SSRs have substantial fitness effects averaging 0.3% per variant. This is the first such estimate that I know of, and, as the authors lay out in the Discussion of practical importance.

There is one element missing from the analysis that I would really like to see if the authors can supply. There is no explicit test, that I can find, of whether the DFEs for SNMs and SSRs is different. The manuscript would be improved by attention to this problem.

COMMENTS FROM THE ACADEMIC EDITOR (lightly edited):

Reviewer #1's advice about the detailed investigation of petite emergence (by learning more about the underlying molecular mechanisms and the randomness/stochasticity of its formation rate) would be indeed an intriguing aspect of the research, but according to [another reviewer], this is out of the scope of the work, which I have to agree with.

Neither the cover letter nor the title/abstract mentions the emerging petites in relation to the fitness effects, so it also appears to me that conducting such a thorough analysis of this phenomenon is not necessary for the current manuscript because it has littl

---

## [Editor Report · Decision Letter 2]

9 May 2024

Dear Mark,

Thank you for your patience while we considered your revised manuscript "Spontaneous single-nucleotide substitutions and microsatellite mutations have distinct distributions of fitness effects" for publication as a Research Article at PLOS Biology. This revised version of your manuscript has been evaluated by the PLOS Biology editors and the Academic Editor.

Based on our Academic Editor's assessment of your revision, we are likely to accept this manuscript for publication, provided you satisfactorily address the following data and other policy-related requests.

IMPORTANT - please attend to the following:

a) As mentioned in my email, unfortunately reviewer #1 was not able to re-review in a timely fashion. However, our Academic Editor kindly agreed to check your responses and revisions. S/he is broadly happy with these, and has no further requests, but I've pasted their comments into the foot of this email for your interest.

b Many thanks for providing the data and code in your OSF deposition. Can you confirm that this is sufficient to generate the Figures? Also, please cite the location of the data clearly in all relevant main and supplementary Figure legends, e.g. “The data and code needed to generate this Figure can be found in https://osf.io/XXXXXXXX"

c) Many thanks for supplying the additional code in Github. However, because Github depositions can be readily changed or deleted, please make a permanent DOI’d copy (e.g. in Zenodo) and provide this URL.

We expect to receive your revised manuscript within two weeks. 

*Published Peer Review History*

*Press*

Sincerely,

Roli

Roland Roberts, PhD

Senior Editor

rroberts@plos.org

PLOS Biology

CODE POLICY

DATA NOT SHOWN?

COMMENTS FROM ACADEMIC EDITOR:

I reviewed the authors' responses to the reviewer's comments and the revised manuscript. I agree that the Authors have put considerable effort into more precisely characterizing the mutations in the MA lines, especially following our specific request. They have also adequately addressed reviewer #1's concerns about the formation and frequency of petites. While acknowledging that the process of petite generation is an intriguing biological question, they have noted it is beyond the scope of their study. Still, by accounting for petites in their analyses, they came up with more accurate estimates than if these had been overlooked.

A key observation from this study is that most spontaneous SNMs in MA lines are deleterious with minimal effects on growth rate – something that would be easy to miss with less sensitive fitness measurement methods. This supports the view that deleterious mutations, often due to hitchhiking, are prevalent in clonal populations and can influence evolutionary trajectories. The new sequencing results also highlight the importance of non-SNM mutations, revealing strains with growth changes despite having no identified SNMs or only harboring typically ignored synonymous/intergenic mutations.

Their further analyses show that mutations in single sequence repeats (SSRs) also impact growth rates. They have further categorized the effects of SSR mutations, finding many with moderate (e.g., missense variants and in-frame deletions and insertions) or high impacts (e.g. frameshift mutations). Reviewer #1 also brought up the issue that most probably other repetitive sequences like LTRs or telomeres/centromeres are likely accumulating mutations that remain undetected with current sequencing technologies. While acknowledging that such mutations could influence growth, the Authors argue convincingly that these are unlikely to account for most of the variation in the selection coefficient, given their lower prevalence in the genome compared to SSRs (that take up to ~10%).

In short, the revised manuscript convincingly argues that most spontaneous mutations accumulating in MA lines are not neutral and usually have a small deleterious effect on growth. This includes mutations in SSR regions, which seem to play an important role in trait variation, even in the absence of non-repeat SNMs. Considering the previous feedback from the other reviewers and the comprehensive revision addressing reviewer #1's concerns, and given the broad potential impact of these findings, I'd recommend acceptance of the manuscript.

---

## [Editor Report · Decision Letter 3]

4 Jun 2024

Dear Mark,

Thank you for the submission of your revised Research Article "Spontaneous single-nucleotide substitutions and microsatellite mutations have distinct distributions of fitness effects" for publication in PLOS Biology. On behalf of my colleagues and the Academic Editor, Csaba Pal, I'm pleased to say that we can in principle accept your manuscript for publication, provided you address any remaining formatting and reporting issues. These will be detailed in an email you should receive within 2-3 business days from our colleagues in the journal operations team; no action is required from you until then. Please note that we will not be able to formally accept your manuscript and schedule it for publication until you have completed any requested changes.

IMPORTANT: Many thanks for providing the data and code in your OSF and Zenodo deposition. I've asked my colleagues to include one remaining request among their own: "Please cite the location of the data clearly in all relevant main and supplementary Figure legends, e.g. “The data and code needed to generate this Figure can be found in https://osf.io/XXXXXXXX and/or https://zenodo.org/records/XXXXXXXX"

Best wishes,

Roli

Senior Editor

PLOS Biology

rroberts@plos.org